



# N₂O changes from the Last Glacial Maximum to the preindustrial - part I:
# quantitative reconstruction of terrestrial and marine emissions using N₂O stable isotopes in ice cores

Hubertus Fischer[1], Jochen Schmitt[1], Michael Bock[1], Barbara Seth[1], Fortunat Joos[1], Renato Spahni[1], Sebastian Lienert[1], Gianna Battaglia[1], Benjamin D. Stocker[1,2], Adrian Schilt[1,3], Edward J. Brook[3]

[1]Climate and Environmental Physics, Physics Institute and Oeschger Centre for Climate Change Research, University of Bern, Bern, CH-3012, Switzerland.
[2]CREAF, E08193 Bellaterra (Cerdanyola del Vallès), Spain
[3]College of Earth, Ocean, and Atmospheric Sciences, Oregon State University, Corvallis, Oregon 97331, USA.

Correspondence to: Hubertus Fischer (hubertus.fischer@climate.unibe.ch)

**Abstract.** Using high precision and centennial resolution ice core information on atmospheric nitrous oxide concentrations and its stable nitrogen and oxygen isotopic composition, we quantitatively reconstruct changes in the terrestrial and marine N₂O emissions over the last 21,000 years. We show that N₂O emissions from land and ocean increased largely in parallel by $1.7 \pm 0.3$ TgN yr$^{-1}$ and $0.7 \pm 0.3$ TgN yr$^{-1}$ over the deglaciation, respectively. However, during the abrupt Northern Hemisphere warmings at the onset of the Bølling/Allerød and the end of the Younger Dryas, terrestrial emissions respond more rapidly to the northward shift in the Intertropical Convergence Zone connected to the resumption of the Atlantic Meridional Overturning Circulation. 90 % of these large step increases were realized within maximum two centuries. In contrast, marine emissions start to slowly increase already many centuries before the rapid warmings, possibly connected to a re-equilibration of subsurface oxygen in response to previous changes. Marine emissions decreased, concomitantly with changes in atmospheric CO₂ and δ¹³C(CO₂), at the onset of the termination and remained minimal during the early phase of Heinrich Stadial 1. During the early Holocene a slow decline in marine N₂O emission of 0.4 TgN yr$^{-1}$ is reconstructed, suggesting an improvement of subsurface water ventilation in line with slowly increasing Atlantic overturning circulation. In the second half of the Holocene total emissions remain on a relatively constant level, however with significant millennial variability which is currently still difficult to attribute to marine or terrestrial sources. Our N₂O emission records provide important quantitative benchmarks for ocean and terrestrial nitrogen cycle models to study the influence of climate on nitrogen turnover on time scales from several decades to glacial/interglacial changes.



## 1 Introduction

Nitrous oxide ($N_2O$) is an important greenhouse gas that contributes to ongoing and past global warming (Stocker et al., 2013;Schilt et al., 2010b) and is involved in the destruction of stratospheric ozone (Myhre et al., 2013). Its past atmospheric variations are recorded in polar ice cores although some sections of some records can be affected by in situ formation of

$N_2O$. $N_2O$ is produced through nitrification and denitrification on land and in the ocean, where nitrification appears to be the dominant pathway in marine environments (Battaglia and Joos, 2017). Total natural net $N_2O$ sources are estimated to amount to $10.5\pm1$ TgN $yr^{-1}$, where about 60 % are estimated to be of terrestrial origin (Battaglia and Joos, 2018a), in line with IPCC AR5 terrestrial emission estimates of 6.6 (3.3 to 9.0) TgN $yr^{-1}$. Atmospheric $N_2O$ is photochemically destroyed in the stratosphere (Ciais et al., 2013) with an estimated preindustrial lifetime of 123 yrs (Prather et al., 2015).

Reconstructions of past variations in terrestrial and marine $N_2O$ emissions from ice core data (Schilt et al., 2014) provide information on the response of the global coupled carbon-nitrogen cycle to past climate variations. Atmospheric $N_2O$ increased from 270 ppb (MacFarling Meure et al., 2006) to around 330 ppb (https://www.esrl.noaa.gov/gmd/) over the industrial period due to human activities such as fertilizer applications (Bouwman et al., 2013;Ciais et al., 2013).

Atmospheric $N_2O$ varied between natural upper and lower bounds of around 300 and 180 ppb, respectively, over glacial-interglacial cycles (Sowers et al., 2003;Spahni et al., 2005;Schilt et al., 2010a). Moreover, atmospheric $N_2O$ concentrations showed a clear positive imprint of rapid climate changes in the North Atlantic region that occurred during the last ice age (so called Dansgaard Oeschger (DO) events) and during the last glacial/interglacial transition (the Bølling/Allerød (B/A) warming and the rapid warming at the end of the Younger Dryas (YD) into the Preboreal (PB)). These rapid warming events

have been shown to be associated with a resumption of the Atlantic Meridional Overturning Circulation (AMOC) (Henry et al., 2016;Pedro et al., 2018;McManus et al., 2004) and a connected northward shift of the Intertropical Convergence Zone (ITCZ) leading to significant changes in monsoon precipitation (Wang et al., 2008). Available $N_2O$ data over the Holocene also show significant albeit smaller longer-term $N_2O$ variability. Until recently it had not been possible to attribute the observed concentration changes to terrestrial or marine sources and ecosystem and physical processes leading to changes in

$N_2O$ production remained not well understood.

Using the distinct nitrogen isotopic signature of terrestrial and marine $N_2O$ sources, high-resolution data on the isotopic composition of $N_2O$ from Greenland and Antarctic ice cores allow us to reconstruct past variations in terrestrial and marine $N_2O$ emissions and, thus, on C-N coupling on time scales from several decades to many centuries. A recent ice core study on

the stable isotope composition of $N_2O$ over the time interval from 16 to 10 thousand years before present (ka BP, where present is defined as 1950) demonstrated the power of such an isotopic approach (Schilt et al., 2014). The isotopic data showed that both land and marine sources increased during the interval from 16 to 10 ka BP and showed a clear imprint on AMOC variations and related climate changes on the terrestrial and marine carbon and nitrogen cycles (Schilt et al., 2014).



Here we attempt a more detailed look at the temporal evolution and phasing of marine and terrestrial N₂O emissions over this time interval. Moreover, no previous study was able yet to reconstruct the complete temporal evolution in terrestrial versus marine emissions of N₂O since the Last Glacial Maximum (LGM; 21 ka BP, including the onset of the termination and the early deglacial "Mystery Interval") and to reconstruct the more subtle changes in the course of the Holocene.

Accordingly, the aim of this study is to improve our understanding of the changes in the main N₂O sources over these time intervals. In this study we reconstruct the emissions of N₂O from land and from the ocean over the entire glacial termination and the Holocene, using new ice core N₂O isotope data. We contrast these emission changes to the accompanying changes in climate parameters and ocean circulation as archived in ice cores and marine sediments. In the accompanying paper (Joos et

al., this issue) these new emission records of the past 21 kyr are used to explore and test alternative mechanisms of the functioning of the C-N cycle on land and to quantify the leading environmental parameters controlling land-based N₂O emissions over this time interval, while the response of the marine nitrogen cycle to rapid changes in AMOC has been recently studied by several coupled biogeochemistry ocean models (Schmittner and Galbraith, 2008;Battaglia et al., 2019;Goldstein et al., 2003).

**2 Methods**

**2.1 Composite records of N₂O, δ¹⁵N(N₂O) and δ¹⁸O(N₂O) from bipolar ice cores**

**2.1.1 New N₂O concentration and dual isotope measurements**

Two hundred two ice core samples were analyzed at the University of Bern for N₂O and its nitrogen and oxygen isotopic composition and compiled with previously published N₂O isotope data measured at Oregon State University over the time

interval 16 to 10 ka BP (Schilt et al., 2014). We measured 103 samples on ice from the North Greenland Ice Core Project (NGRIP), 42 from the the European Project for Ice Coring in Antarctica (EPICA), Dronning Maud Land core (EDML), 57 from the Talos Dome Ice Core (TALDICE). 13 samples of these 202 samples are considered to be affected by in situ production of N₂O (7 for NGRIP, 5 for EDML, and 1 for TALDICE) and were excluded from our data set (see details on outlier detection below).

The new data cover the period from about 27 to 0.3 ka BP and allow us to reconstruct for the first time the N₂O isotopic composition for the LGM, the early deglacial period and for the entire Holocene (last 11 kyr). Two independent analytical systems were used at the University of Bern for both δ¹⁵N(N₂O) and δ¹⁸O(N₂O) as described in detail previously (Bock et al., 2014;Schmitt et al., 2014). δ¹⁵N(N₂O) data are given with respect to atmospheric N₂, δ¹⁸O(N₂O) data are reported with

respect to VSMOW (Vienna Standard Mean Ocean Water), both in the commonly used delta notation. Our reference is a recent air sample, NAT332, measured at Utrecht University (Sapart et al., 2011) and cross-referenced to Air Controlé, an in-



house standard in Bern (Schmitt et al., 2014). Finally, all new Bern data presented here are shifted by small, constant offsets (-0.80 ‰ for $\delta^{15}N(N_2O)$ and +0.36 ‰ for $\delta^{18}O(N_2O)$) to account for laboratory differences among results measured at the University of Bern and at Oregon State University as described previously (Schilt et al., 2014).

The analytical precision for the two systems used at the University of Bern and different sample batches is given in Table 1. Error bars of all isotopic data in this publication represent one standard deviation ($\pm 1\sigma$) of ice sample replicates or are based on the reproducibility of air reference measurements used to calibrate ice samples. Measurement precisions of individual samples for $N_2O$, $\delta^{15}N(N_2O)$ and $\delta^{18}O(N_2O)$ are always better than ± 5 ppb, ± 0.4 ‰ and ± 0.8 ‰, respectively (Table 1).

### 2.1.2 The composite ice core records – data and age scale

Using measurements performed on various Greenland and Antarctic ice cores, composite records for $N_2O$ concentration and dual isotope signatures were derived over the last up to 28 kyr (Fig. 1). To this end our new $N_2O$ concentration data were combined with published $N_2O$ concentration data from NorthGRIP (Schilt et al., 2010b) after exclusion of data points likely affected by in situ $N_2O$ formation (see below), EPICA Dome C (EDC) (Flückiger et al., 2002;Spahni et al., 2005)), and Taylor Glacier ice (Schilt et al., 2014). The EDC data included are for ages younger than 14.5 ka BP and the published

Taylor Glacier data (Schilt et al., 2014) cover the period from 15.9 to 9.9 ka BP. We do not include published $N_2O$ concentration data from previous Talos Dome measurements (Schilt et al., 2010b) in the composite due to unresolved analytical offsets between these published and our new $N_2O$ concentration data. The new $N_2O$ double isotope data are combined with published $N_2O$ double isotope data from Taylor Glacier (Schilt et al., 2014) measured at Oregon State University covering the period 15.9 to 9.9 ka BP; the analytical procedures at Oregon and Bern are tightly cross-calibrated

(Schilt et al., 2014;Schmitt et al., 2014).

In order to compare $N_2O$ data from different ice records, they must be put on a common age scale. We start from the existing AICC2012 age scale (Veres et al., 2013) and compare the concentrations of the fast varying and globally well mixed greenhouse gas methane ($CH_4$) between ice cores (Fig. 2). In the process of establishing the AICC2012 age scale, Greenland

and Antarctic ice cores were synchronized by aligning rapid climate changes in Greenland $\delta^{18}O$ with rapid $CH_4$ variations in Antarctic ice cores (Veres et al., 2013). Unfortunately, due to the ice age-gas age difference between Greenland $\delta^{18}O$ and $CH_4$ records, this leaves some uncertainty when comparing Greenland and Antarctic gas records. Accordingly, new high-resolution $CH_4$ data reveal that AICC2012 gas ages are offset by several 100 years at the start of the B/A and before and after the YD (Fig. 2). This is also true for other time periods during the last glacial period (Baumgartner et al., 2014), where rapid

$CH_4$ increases occurred. To correct for this age offset, we match mid points of fast $CH_4$ variations (B/A and YD) seen in all our ice cores to the fast climatic changes indicated by $\delta^{18}O_{ice}$ at NGRIP on the absolutely counted GICC05 age scale



(Rasmussen et al., 2006). This assumes, that the CH₄ response is synchronous to rapid climate changes within a few decades as previously shown for the NGRIP ice core (Baumgartner et al., 2014).

A well-known problem with $N_2O$ data is that they can be subject to in situ $N_2O$ formation for selected ice core sections. In situ production leads to elevated $N_2O$ concentrations and abnormal isotopic signatures compared to other, coeval, ice core records. For Greenland ice, in situ artefacts typically occur as erratic $N_2O$ outliers during the glacial, especially when the ice chemistry changes during major climatic transitions. (see Fig. 1; measurements with identified in situ production are marked by red circles). For several Antarctic ice cores (Schilt et al., 2010b;Flückiger et al., 2004;Flückiger et al., 1999), the dust-rich LGM sections are almost entirely affected by in situ production (see EDC and EDML records in Fig. 1). Due to its lower impurity content, the TALDICE ice core as well as the outcropping ice from Taylor Glacier (Schilt et al., 2014) appears to be an exception and potential artefacts have been found only in very few instances (Schilt et al., 2010b).

The most straightforward criterion to identify samples affected by in situ production targets anomalies in the $N_2O$ concentration. The assumption is that $N_2O$ cannot be consumed or lost in the ice, but only produced. An outlier detection method was for example used for the highest resolution $N_2O$ record available to date (both in sample frequency and in terms of the width of the bubble enclosure characteristics) from the Greenland NGRIP ice core (Schilt et al., 2013) covering the time interval from 110-10 ka BP. Individual samples in this record were removed if they exceeded a threshold of 8 ppb above a spline approximation through all samples (Flückiger et al., 2004). This outlier algorithm is not well suited to detect in situ formation in times when $N_2O$ concentrations increase strongly due to rapid emission increases. Thus, the outlier detection may fail during the onset and end of DO events. In fact, as illustrated in Fig. 2, the published NGRIP $N_2O$ record appears to be systematically higher than its Antarctic counterparts from the Talos Dome ice core and from Taylor Glacier after the onset of the B/A and after the end of the YD, although the long atmospheric lifetime of $N_2O$ requires a very small interhemispheric gradient in the range of the measurement uncertainty. The higher NGRIP concentrations in these intervals can also not be attributed to a more narrow bubble enclosure characteristics, thus better resolution of the record compared to the Antarctic data, as the concurrent increase in atmospheric CH₄ concentrations is recorded equally fast in the NGRIP and Taylor Glacier data (Fig. 2). Accordingly, we cannot exclude an in situ contribution to the rapid $N_2O$ increases in the NGRIP ice core and excluded the respective samples (red symbols in Fig. 2) from our data set compiled from all cores.

To obtain a final data set compiled from all cores after the pre-screening described above and rigorously remove samples that may still be affected by in situ processes, individual samples were finally excluded in our study from the data set compiled from all ice cores if they were at least 20 ppb higher than the average of our Monte Carlo smoothing spline (see details on our Monte Carlo Average (MCA) below). For a group of EDML isotope measurements (22 - 18 ka BP) this MCA method could not be applied, because precise $N_2O$ concentration measurements were not available for these samples. These samples were, therefore, excluded as in situ samples as well, as all surrounding EDML samples from this interval showed





elevated N$_2$O concentrations (Fig. 1). The final data set after age scale synchronization and outlier removal is shown in Fig. 3.

As none of the isotope and concentration records is continuous and as isotope measurements are only available at a lower
resolution, we derived a common lower frequency record using a spline fitting routine (Enting, 1987), which served as input into the box model inversion used to calculate emissions as described below. As the time interval younger than 16 ka BP was better resolved than the glacial part of the record, we used a smaller cutoff period (higher frequency content) of the spline for ages < 16 ka BP and a larger cutoff period for older ages. Thus, the time interval before 16 ka BP is more strongly low-pass filtered than that after 16 ka BP. To test the sensitivity of the choice of cutoff period on our emission reconstruction, we also
used splines with different but constant cutoff periods throughout the record (not shown). A higher frequency content (equivalent to a cutoff period of 900 years or lower) prior to 16 ka BP leads to erroneous variations in the spline approximation that are not supported by the measured $\delta^{15}$N(N$_2$O) data and consequently to erroneous variability in terrestrial and marine emissions. The same holds true for cutoff periods lower than 500 years in the time interval after 16 ky BP. Vice versa, very long cutoff periods (longer than 900 years) in the record younger than 16 ka BP lead to a substantial
underestimation of the N$_2$O and $\delta^{15}$N(N$_2$O) variability connected to the oscillations accompanying the B/A warming and the YD cold swing. Accordingly, for our best reconstruction scenario we used a cutoff period of 2000 years prior to 16 ka BP and 700 years thereafter. This change in frequency content at 16 ka BP has to be taken into account when interpreting the data, however, the majority of the centennial to millennial N$_2$O emission changes discussed below is found in the time interval after 16 ka BP, where a consistent cutoff period of 700 years was used.

To derive the uncertainty of the spline approximation representative for the time scale resolved by the low-pass filter, the spline fitting was repeated 1,000 times using a Monte Carlo approach, where the measurements were varied randomly within their analytical uncertainties. This Monte Carlo spline and its uncertainty was then used to detect outliers affected by in situ formation as described above. The procedure was iterated until no more outliers were found and the final Monte Carlo spline
was calculated using the final data set (Fig. 3).

To assess the accuracy of ice core N$_2$O concentrations for recent times, we compare the ice core data with overlapping firn air and direct atmospheric air measurements (Fig. 4). Reconstructions from the high-accumulation Law Dome drill site offer the best connection with the current atmospheric measurements (Etheridge et al., 1988;Francey et al., 1999). The Law Dome
ice core record covers the time interval from 2 ka BP to 1979 CE and air samples obtained from the firn column and the Cape Grim air archive cover 1942 CE to 2004 CE (MacFarling Meure et al., 2006) (Fig. 4). Our new measurements using NGRIP, EDML and TALDICE ice core samples and published records from EDC and EDML overlap within the two standard deviation envelope of a Monte Carlo average spline through the Law Dome and Cape Grim data with a cutoff period of 250 yr (Fig. 3). Our new concentration data also generally agree with a new ice core data set based on the

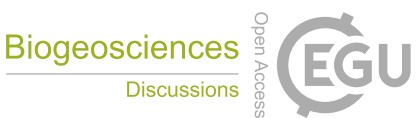

Greenland NEEM ice core (Prokopiou et al., 2018), which however shows negative outliers compared to our record. These are especially pronounced in NEEM ice core samples prior to 2000 BP with $N_2O$ concentrations as low as 240 ppb. Note that the data by Prokopiou et al. (2018) are based on a dry extration technique which does not ensure 100 % extraction efficiency. Moreover, the extraction efficiency for $N_2O$ in air bubbles, clathrates or $N_2O$ dissolved in the ice lattice may be different and

could potentially be related to the low $N_2O$ concentrations in this interval. Our new compilation based on Greenland (NGRIP) and Antarctic (EDC, EDML, Talos Dome) ice cores consistently shows concentrations of 265-270 ppb in this time interval, similar to what is found during preindustrial times. In view of the good agreement of several Greenland and Antarctic ice cores and the better data resolution in our record, we decided not to use the NEEM samples by Prokopiou et al. (2018).

In case of the $N_2O$ isotopic records very few other reconstructions are available. NGRIP results (Bernard et al., 2006) connect our youngest data points with the firn air and Cape Grim archive measurements (Fig. 3) capturing the recent anthropogenic perturbation (Park et al., 2012). The recent study by Prokopiou et al. (2018) also provides $\delta^{15}N(N_2O)$ and $\delta^{18}O(N_2O)$ data prior to the preindustrial period from the NEEM ice core (Fig. 3). Isotopic analyses in that publication also

differentiate the nitrogen isotopic signature of molecules with the $^{15}N$ located at the central $\delta^{15}N^{\alpha}(N_2O)$ or terminal $\delta^{15}N^{\beta}(N_2O)$ position, which we cannot distinguish by our method.

The $\delta^{15}N(N_2O)$ data from the NEEM ice core (Prokopiou et al., 2018) are in general agreement with our data set measured at the University of Bern based on Greenland (NGRIP) and Antarctic (EDML, Talos Dome) ice cores. However, our data show

rather constant nitrogen isotopic signatures around +9 ‰ to +10 ‰ over the time interval 3.3-0.2 ka BP, i.e., significantly less scatter than the data by Prokopiou et al. (2018). This overall supports the conclusions drawn from the $\delta^{15}N(N_2O)$ record for the late Holocene (Prokopiou et al., 2018), but our more precise data constrain the late Holocene $N_2O$ budget and its variability much more closely. Accordingly, we refrain from using the $\delta^{15}N(N_2O)$ data by Prokopiou et al. (2018) in our model deconvolution.

Also our $\delta^{18}O(N_2O)$ data show rather constant values in this interval between +45 ‰ and + 46 ‰ in this interval. The values of Prokopiou et al. (2018) are higher in this time interval by +1 ‰ to up to +6 ‰ relative to our data. Such large offsets cannot be explained by differences in the isotopic standards used in the different labs. The much larger variability in the data by Prokopiou et al. (2018) connected to generally higher $\delta^{18}O(N_2O)$ compared to our data is not easy to explain. Any

fractionation occurring during an (incomplete) gas extraction is expected to have an effect on $\delta^{15}N(N_2O)$ as well, which is not observed. Incomplete cryo-sampling of $N_2O$ during gas collection may explain lower $N_2O$ concentrations and higher $\delta^{18}O(N_2O)$ but again a similar effect on $\delta^{15}N(N_2O)$ would be expected. In view of the better data quality of our record we refrain from using the data by Prokopiou et al. (2018) in our $\delta^{18}O(N_2O)$ compilation. As we do not use $\delta^{18}O(N_2O)$ in our box




model deconvolution to constrain terrestrial and marine emissions, the offset between the two data sets has no impact on the conclusions drawn in our study.

## 2.2 Reconstructing marine and terrestrial $N_2O$ emissions by deconvolving the $N_2O$ and $\delta^{15}N(N_2O)$ ice core records

### 2.2.1 Monte Carlo two-box model

We deconvolved the evolution of tropospheric $N_2O$ and $\delta^{15}N(N_2O)$ with a two-box atmosphere model following Schilt et al. (2014). This yields both the global terrestrial and the global marine $N_2O$ source evolution. This novel source reconstruction for the past 21 kyr permits us to disentangle various hypotheses put forward to explain the observed $N_2O$ changes and to test whether nitrogen availability in terrestrial ecosystems can limit nitrogen turnover, hence $N_2O$ production, in soils (Joos et al, this issue). In addition, the reconstructed emission time series serve as benchmark for mechanistic and coupled
climate/biogeochemical models simulating land and ocean $N_2O$ emissions.

The minimal two-box atmosphere model features a tropospheric and stratospheric box as well as a land and ocean source and includes $N_2O$, $\delta^{15}N(N_2O)$, and $\delta^{18}O(N_2O)$ as tracers. Air exchange between the troposphere and stratosphere is assumed time-invariant and $N_2O$ is destroyed in the stratosphere assuming an e-folding lifetime and considering isotopic fractionation.
$N_2O$ sources are inversely calculated by prescribing tropospheric $N_2O$ and $\delta^{15}N(N_2O)$ from the splines of the composite ice records as input data. The two budget equations for tropospheric $N_2O$ and $\delta^{15}N(N_2O)$ are solved at each time step for the two unknown global source fluxes for a given set of model parameters, initial conditions and tropospheric input data.

$\delta^{18}O(N_2O)$ serves as an independent check of consistency. To this end we used the terrestrial and marine emission records to
calculate the tropospheric $\delta^{18}O(N_2O)$ signature in a forward model mode and compared it with the measured $\delta^{18}O(N_2O)$ record. The envelope of $\pm$ 2 standard deviations around the mean of this Monte Carlo forward output is relatively large (Fig. 3) due to the large uncertainties in the source signatures of terrestrial and marine sources. Nevertheless, Fig. 3 shows that our source deconvolution is in agreement with the $\delta^{18}O(N_2O)$ measurements.

A Monte Carlo approach is used in the deconvolution to estimate uncertainties in $N_2O$ land and ocean emissions (see details in Schilt et al. (2014)). Initial conditions and model parameters (Table 2) and tropospheric input data are varied randomly within their uncertainties. Model parameters are kept time invariant during a single Monte Carlo run, except in sensitivity simulations (see below). Model parameters include atmospheric lifetime, exchange rate of air between troposphere and stratosphere, stratospheric fractionation coefficients, and characteristic isotopic compositions of the terrestrial and marine
sources. Most notably, $\delta^{15}N(N_2O)$ is varied between -34 and +2 ‰ for the global terrestrial emissions (Schilt et al., 2014) and between +4 ‰ and +10 ‰  for global marine emissions (Frame et al., 2014). In order to constrain the model by the current knowledge of the ratio of natural marine and land emissions (Battaglia and Joos, 2018a), the allowed fraction of



marine emissions on total emissions ($f_m$) at the start of each run at 28 ka BP must lie between 30 % and 58 % close to the confidence interval for $f_m$ given by Battaglia and Joos (2018a). In principle the model could freely evolve from this starting point, but the results show (see Fig. 7 below) that $f_m$ is not varying largely over time and that the late Holocene value in our runs is very close to the estimate by Battaglia and Joos (2018a). For each Monte Carlo iteration the model is forced with an

individual Monte Carlo realization of the splined ice core data with cutoff periods of 2000 yr (28-16 ka BP) and 700 yr (16-0 ka BP), where the measured values were varied within the distribution of the measurement uncertainty. Average sources and their $2\sigma$ uncertainty bands are then obtained from 500 solutions.

### 2.2.2 Model performance tests and limitations

In order to test whether the model is able to reliably reconstruct terrestrial and marine emissions during rapid changes in the

$N_2O$ budget using $N_2O$ concentration and $\delta^{15}N(N_2O)$ records from ice cores, we used artificial data mimicking the fast emission changes seen at the onset of the BA. These tests become especially important as the true data resolution is limited and because we use a spline approximation as input to the model. Moreover, ice core values represent already a low pass filtered version of the atmospheric record due to the slow bubble enclosure process at the firn/ice transition.

Due to the slightly different atmospheric lifetimes of $^{14}N_2O$ and $^{15}N_2O$, significant variations in the isotopic signature of atmospheric $N_2O$ are expected over a time interval of a few centuries following a rapid increase in $N_2O$ emissions, even if the isotopic signature of the emissions remained constant. Accordingly, our transient deconvolution technique must be also able to correct for this so called "disequilibrium effect" in the low-pass filtered ice core record to derive unbiased terrestrial and marine emissions values.

We performed three test runs assuming (i) a ramp-up of land emissions only over 50 years with a constant nitrogen isotopic signature of -18 ‰, (ii) a ramp-up of marine emissions only over 50 years with a constant nitrogen isotopic signature of +7 ‰, and (iii) a proportional ramp-up of both land and marine emissions over 50 years using the constant isotope signatures given above (Fig. 5). The increase in total emissions was kept the same in all three runs. The emissions were used to drive a

forward version of our model to calculate atmospheric $N_2O$ concentrations as well as its isotopic signatures. We refer to this data set further on as "artificial atmospheric data". The emission changes led to an increase of atmospheric $N_2O$ from about 220 to 260 ppb, similar to what is observed at the B/A onset (Fig. 5). The $\delta^{15}N(N_2O)$ signature of the atmosphere changed from a value of 10 ‰ before the emission change to a value of 12 ‰ in case (i), 8 ‰ in case (ii) and remained the same after fading out of the disequilibrium effect in case (iii) (Fig. 5). These atmospheric records were then low-pass filtered using a

log-normal gas age distribution with a mean age of 132 yr (Köhler et al., 2011) to mimic the bubble enclosure process. According to latest experimental results this represents an upper limit of the width of the age distribution in the bubbles over the deglacial for all the cores used in our compilation (Fourteau et al., 2017). Accordingly, this filter provides a conservative estimate of the maximum effect of the bubble enclosure on our deconvolution. We refer to this low-pass filtered version



further on as "artificial ice core data". The low-pass filtered atmospheric data were then used as input in our Monte Carlo box model deconvolution, where the input data from the artificial ice core data is splined with a cutoff period of 700 years. We refer to this record further on as "splined artificial ice core data". The reconstructed emissions were compared to the input data and contrasted to the reconstructed emission from the true ice core record at the onset of the BA. For the latter

comparison we synchronized the well-defined onset of the CH₄ increase in the ice core record with the start of the N₂O emission increase in our test runs assuming that CH₄ and N₂O started to increase synchronously in response to the rapid warming events.

The results of these performance tests are summarized in Fig. 5 with the increase in N₂O emissions starting at model year

5000. As illustrated in Fig. 5, the model deconvolution is able to unambiguously separate terrestrial and marine emissions despite the low-pass filtering by the bubble enclosure process and the spline approximation and despite the disequilibrium effect. The model is able to reconstruct the total amplitude in emission fluxes well, however, the temporal resolution of fast emission changes is hampered by the low-pass filtering of the data. We also performed tests with a much larger expected value of the gas age distribution of 330 yr (a typical deglacial value derived from firnification models for low accumulation

sites such as EDC (Köhler et al., 2011)) and our method was also able to unambiguously reconstruct the emission changes in this case (not shown).

As can be seen in Fig. 5B and C, the N₂O concentrations in the artificial ice core data and its isotopic composition start to increase a few decades after the onset of the true emission increase in the input data. This can be readily explained by the

log-normal age distribution assumed to mimic the low-pass filter of the bubble enclosure process, which has a very long tail of air bubbles containing very old air, thus leading to a small delay in the onset of the reconstructed emission changes. This effect is more pronounced assuming a mean age of the gas age distribution of 330 yr instead of 132 yr (not shown). In real ice core data this offset is compensated by a shift in the gas age scale relative to the ice age scale.

The long tail of old air in the gas age distribution together with the spline approximation is also the reason for the time needed in the emission reconstruction based on the splined artificial ice core data to reach its maximum, which takes several hundred years longer than the 50 yr ramp-up in the originally assumed emission increase. Note, that a log-normal age distribution is overestimating the width of the true age distribution in the ice and especially the long tail of old air is not realistic. The effect of this low-pass filtering on the synthetic data is, therefore, stronger than the effect of the true bubble

enclosure in the ice and the effects illustrated in Fig, 5 can be regarded as an upper limit of the true delay and the response time. Note also that the reconstructed emission increases in Fig. 5D appear to start about two centuries earlier than the major increase in the input fluxes and the artificial ice core data. However, this lead is an artefact created by the use of the splined artificial ice core data as input to our model.





In view of the reasonable model performance described above, the comparison of the artificial data runs with the true ice core reconstructions for the B/A warming in Fig. 5 suggests that for example the rapid $N_2O$ increase at the onset of the B/A is mainly of terrestrial origin while the major marine emission increase is delayed to this onset (see results and discussion) by a few centuries. While our transient box model deconvolution accounts for the long atmospheric lifetime of $N_2O$, it - of

course - cannot correct for the low-pass filtering of the bubble enclosure process and the spline approximation. This is the reason why the emission reconstruction takes about 350 years from the start of the emission increase in the input data (and about 550 years from the start of the first increase in the reconstructed emissions) to accomplish 90 % of the emission increase and very likely also the reason why the duration of the terrestrial emission increase in the ice core based reconstruction (grey line in Fig. 5) has a similar time scale, while the rapid warming in Greenland and the North Atlantic

occurred only in a few decades (Steffensen et al., 2008;Erhardt et al., 2018).

In summary, our model can reliably separate terrestrial and marine emissions. The onset of rapid increases in total $N_2O$ emission can be clearly identified using the high resolution $N_2O$ concentration record itself, however, the resolution of individual terrestrial and marine emission reconstructions is still limited by the available resolution of the $\delta^{15}N(N_2O)$ record.

The precise temporal evolution of emission changes could be improved in the future if the true gas age distribution in the ice were known (Fourteau et al., 2017) and when even higher data resolution and precision become available that allow us to use input data with a higher frequency content than our spline with a cutoff period of 700 years.

### 2.2.3 Sensitivity runs

In two sensitivity tests, we investigated the influence of the prescribed initial range of the marine contributions to total $N_2O$

emissions and varied the initial fraction of marine emissions only between 25 % and 35 % and between 53 and 63 % as alternatives to our standard run, where this contribution was varied over a larger range from 30 % to 58 % (Fig. 6). Sensitivity simulations were also used to probe the influence on time-varying changes in the isotopic signature of terrestrial (Amundson et al., 2003;McLauchlan et al., 2013) and marine $N_2O$ emissions (Galbraith and Kienast, 2013) and in the atmospheric lifetime (Kracher et al., 2016).

$\delta^{15}N(N_2O)$ (as well as $\delta^{18}O(N_2O)$) of terrestrial and marine $N_2O$ emissions are kept time-invariant in the standard deconvolution setup, however, these isotopic source signatures may have changed over time. We constructed scenarios for temporal changes in the $\delta^{15}N$ signatures of global terrestrial and marine $N_2O$ emissions to test the sensitivity of inferred terrestrial and ocean $N_2O$ emissions to these changes in source signatures (Fig. 6). The scenarios are based on scarce, and

partly conflicting, observational evidence. They are used here solely as indicative measures of the potential magnitude of plausible temporal changes in the $\delta^{15}N$ signature of emitted $N_2O$.



Amundson et al. (2003) provide empirical relationships between $\delta^{15}$N of the integrated soil N pool (0-50 cm depth), $\delta^{15}$N$_{soil}$, and local annual mean temperature, MAT, and annual mean precipitation, MAP:

$$\delta^{15}\mathrm{N}_{soil} = 4.3266\ ‰ + 0.2048\ ‰°\mathrm{C}^{-1} \cdot \mathrm{MAT} - 0.0012\ ‰\mathrm{mm}^{-1} \cdot \mathrm{MAP}$$

This relationship is derived from modern spatial gradients in climate and soil nitrogen isotopic signatures. Here, we transferred this relationship to estimate the potential magnitude of a temporal change in $\delta^{15}$N of N$_2$O emitted from soils. The relationship was evaluated for each grid cell using transient climate data from the TraCE-21kyr model run covering the last 21,000 years (Liu et al., 2009;Otto-Bliesner et al., 2014). Global average signatures are calculated as the flux weighted mean

of simulated N$_2$O emissions in LPX-Bern (version 1.4) at decadal resolution and for the past 21,000 years (see Joos et al, this issue for details of the setup of LPX-Bern). Similarly, soil experiments of N$_2$O production reveal that the difference of $\delta^{18}$O between soil water and produced N$_2$O is linearly related to the water filled pore space (WFPS) and we applied the relationship given by Lewicka-Szczebak et al. (2014) using WFPS and N2O emissions as simulated by LPX-Bern (version 1.4). This yields an N$_2$O emission-weighted, global mean change in isotopic source signature from 18 to 0 ka BP of +0.4 ‰

for $\delta^{15}$N and of +0.8 ‰ for $\delta^{18}$O (dark green line in Fig. 6A and B). The reliability of these estimates remains questionable as it is, for example, not clear whether the current geographical relationship between bulk soil $\delta^{15}$N with climate holds also for temporal changes in $\delta^{15}$N of emitted N$_2$O from soils.

$\delta^{15}$N data of lacustrine sediments (McLauchlan et al., 2013) suggest an opposite signal than estimated from the empirical

climate-$\delta^{15}$N relationship. The stacked data show a 2 ‰ decrease in $\delta^{15}$N from 15 to 7 ka BP and an 0.6 ‰ increase thereafter (light green line in Fig. 6B). The stack of lacustrine sediments is from sites predominantly outside the main N$_2$O emission regions (see Fig. 1 in McLauchlan et al. (2013)), and may not be representative for global changes in $\delta^{15}$N in soils. In addition, it is plausible that with a change in soil N availability, pathways of N loss and fractionation processes relevant for N$_2$O emissions may change as well. In brief, the $\delta^{15}$N trend from lacustrine sediments may not necessarily correspond to

the trend in $\delta^{15}$N of N$_2$O emitted from the land biosphere.

As a measure of potential changes in $\delta^{15}$N of marine N$_2$O emissions, we apply reconstructed changes in $\delta^{15}$N of marine sediment material (blue line in Fig. 6B) which amount to about +0.4 ‰ from 18 to 0 ka BP period (Galbraith and Kienast, 2013). Further, the trend in $\delta^{18}$O from a global benthic isotope stacks corrected for temperature changes (Elderfield et al.,

2012;Lisiecki and Raymo, 2005) is applied to estimate potential temporal changes in $\delta^{18}$O(N$_2$O) (blue line in Fig. 6A).





This yields two scenarios for isotopic source change. The temporal changes in $\delta^{15}$N of global terrestrial emissions are either prescribed according to the empirical relationship (scenario 1; purple line in 6C to E) or from changes in $\delta^{15}$N from stacked records of lacustrine sediments (scenario 2; red line in 6C to E). In both scenarios marine $\delta^{15}$N and $\delta^{18}$O source signatures are varied according to the evidence from marine sediment records. The resulting land and ocean $N_2O$ emissions are shown in Fig. 6C and 6D.

Finally, we tested the sensitivity of our box model approach to temporal changes in the atmospheric lifetime (yellow lines in Fig. 6C-E), which is controlled by the troposphere/stratosphere exchange. The Brewer-Dobson circulation and atmospheric lifetime of $N_2O$ may change over time. Kracher et al. (2016) find a linear relationship between the relative change in atmospheric lifetime of $N_2O$ and the change in global mean surface air temperature (SAT) with a slope of -5.87 %/K in idealized global warming simulations. Taken at face value, their results imply a longer lifetime during LGM and a shorter lifetime in the early Holocene compared to preindustrial conditions. We applied the slope of Kracher et al. in a simulation with an idealized deglacial temperature evolution. SAT was assumed to increase linearly by 3.5°C between 17 ka BP and 11 ka BP and to decrease linearly by 0.7°C over the Holocene (Marcott et al., 2013;Shakun et al., 2012); the $N_2O$ lifetime was varied accordingly. Obviously, the box model does not represent the Brewer-Dobson Circulation, but summarizes troposphere-to-stratosphere exchange in a single air mass exchange parameter only. Therefore, we varied the stratospheric decomposition rate of $N_2O$ in the model to enforce changes in lifetime. The scenario yields an atmospheric lifetime that is about 16 % (20 yr) higher for LGM than for preindustrial conditions. Accordingly, the inferred marine and terrestrial emissions over the past 21 kyr are about 16 % higher than in the standard deconvolution setup (Fig. 6B and C). Within the framework of our two-box model, the partitioning between marine and terrestrial emissions is not sensitive to this temporal change in lifetime.

## 3 Results

### 3.1 Ice core data of $N_2O$, $\delta^{15}$N($N_2O$) and $\delta^{18}$O($N_2O$)

Reconstructed tropospheric $N_2O$ shows an overall increase from around 210 ppb at the Last Glacial Maximum (LGM, 21 ka BP) to 270 ppb before the onset of industrialization (Fig. 3). During the late glacial, $N_2O$ concentrations alternated around a relatively constant level of around 210 ppb after declining from an elevated level of up to 240 ppb during DO event 3 at around 28.5 ka BP (see Fig. 1). Even during this late glacial interval between 26-18 ka BP, significant variations in $N_2O$ concentrations can be resolved with pronounced concentration minima of around 200 ppb at 25 ka BP (potentially related to Heinrich event 2), at 24.5 ka BP and less pronounced at around 23.5 ka BP.





During the deglacial, the N₂O record features a pronounced millennial scale decrease by about 15 ppb during 17.4 to 16.6 ka in the early phase of the Heinrich Stadial 1 Northern Hemisphere (NH) cold phase (HS1; 17.4 to 14.6 ka BP (Rasmussen et al., 2014)) followed by a slow recovery in the second phase of HS1. At the end of HS1 (the onset of the B/A NH warm period; 14.6 to 12.8 ka BP) N₂O shows a very rapid increase by almost 40 ppb, followed by a slower increase to peak values

in the late B/A. N₂O decreased again by about 20 ppb during 13 to 12 ka BP in the early part of the YD NH cold period (YD; 12.8 to 11.7 ka BP), followed by a recovery during the late YD to reach peak values around 11 ka BP after a rapid increase at the end of the YD.

During the early Holocene N₂O shows another decrease from about 270 to 260 ppb in the interval from 11 to 10 ka BP

followed by rather constant mixing ratios until about 5 ka BP. After 5 ka BP N₂O increases to about 270 ppb and is generally more variable than in the early Holocene. In this period, N₂O varies on multi-centennial to millennial time scales between 272 and 262 ppb with maxima around 4.5 ka BP, 2.6 ka BP and minima at 3.8 ka BP and 1.4 ka BP. Especially for the most recent millennia (Fig. 4) there exists excellent overlap of N₂O data from many different ice core sites and analytical techniques and the ice core record is seamlessly linked to instrumental N₂O measurements of tropospheric background air by

the Law Dome data (MacFarling Meure et al., 2006). This not only provides confidence in the ice core reconstructions, but also provides a robust view on the millennial scale N₂O oscillation during the Late Holocene. From the local N₂O maximum with ca. 270 ppb at around 2.6 ka BP, N₂O dropped by around 8 ppb to reach a minimum at 1.4 ka BP followed by a recovery within around 200 to 400 years (Fig. 4).

δ¹⁵N(N₂O) varied within only about 2 ‰ over the past 28 kyr (Fig. 1). This suggests that the relative contribution of isotopically light and isotopically heavy emission sources did not change substantially. There is an overall net change in δ¹⁵N(N₂O) from 22 ka BP to the preindustrial by about -1 ‰ and some small but distinct variations in the course of the last glacial termination. Prior to 22 ka BP, δ¹⁵N(N₂O) data exist only from the Greenland NGRIP ice core (Fig. 1), which appears to be isotopically quite low in δ¹⁵N(N₂O) for dusty glacial ice compared to the Antarctic Talos Dome (TALDICE) ice core,

where data exist after 23 ka BP. As the NGRIP data showed to be more prone to in situ N₂O production compared to the TALDICE core, we refrain from quantitatively interpreting the N₂O isotope data prior to 21 ka BP using the box model deconvolution. During the time period from 21 ka BP to the preindustrial (Fig. 3) it is interesting to note that the minima in N₂O around 17.5 ka in HS1 seems reflected by a corresponding minimum in δ¹⁵N(N₂O), whereas the two other distinct minima in δ¹⁵N(N₂O) at 14.6 ka BP and 11.6 ka BP coincide with the rapid N₂O rise at the onset of the B/A and the

Preboreal NH warm phases. In the two latter cases atmospheric δ¹⁵N(N₂O) is highly affected by the disequilibrium effect and a model deconvolution is required to reliably derive the dynamics in nitrous oxide source changes. The overall decrease in δ¹⁵N(N₂O) of about 1 ‰ from the LGM to the preindustrial is not affected by this effect and points to an increasing share of terrestrial emissions on total emissions over the deglaciation.



$\delta^{18}O(N_2O)$ also varied modestly (typical range of 3 ‰) during the past 28 kyr around a mean value of about 45 ‰ (Fig. 1). Again, we concentrate on the interpretation of the last 21 kyr (Fig. 3) in terms of isotope changes as the $\delta^{18}O(N_2O)$ signature of the NGRIP data prior to 23 ka BP appears to be offset (by about +1 ‰) relative to the TALDICE data. The $\delta^{18}O(N_2O)$

record over the last 21 kyr suggests a minimum around 15.5 ka BP and highest values during the B/A, while $\delta^{18}O(N_2O)$ varied around 45 to 46 ‰ in the Holocene and comparable to LGM values. Despite a lower signal to noise ratio compared to $\delta^{15}N(N_2O)$, $\delta^{18}O(N_2O)$ appears to become progressively isotopically lighter (more negative) in the first half of the Holocene and seems to increase again thereafter. Note that there is no overall correlation between the $\delta^{15}N(N_2O)$ and $\delta^{18}O(N_2O)$ records. The low correlation may be linked to overall small variability in the two records and to differences in the processes

affecting $\delta^{15}N(N_2O)$ and $\delta^{18}O(N_2O)$ with $\delta^{18}O$ also being affected by changes in the water cycle and ice sheet mass, while $\delta^{15}N(N_2O)$ being dependent on the efficiency of nitrogen turnover in soils or in low oxygen water masses of the ocean.

### 3.2 Reconstructed changes in terrestrial and marine $N_2O$ emissions: deconvolving the ice core $N_2O$ and $\delta^{15}N(N_2O)$ records

#### 3.2.1 Inferred emission changes

The $N_2O$ and $\delta^{15}N(N_2O)$ records allow us to disentangle changes in global terrestrial and global marine $N_2O$ emissions to the atmosphere over the last 21 kyr. To this end we used the two-box model deconvolution of the $N_2O$ and $\delta^{15}N(N_2O)$ records described in section 2.2 to determine marine and terrestrial $N_2O$ emissions and their uncertainty (Fig. 7). $\delta^{18}O(N_2O)$ was not used as additional constraint in the deconvolution but was used as independent check in forward modeling of the atmospheric concentrations and isotopic signature from the reconstructed emissions (Fig. 3). Reconstructed terrestrial and

marine $N_2O$ emissions increased between the LGM (21 ka BP) and full interglacial conditions (7 ka BP) by $1.5 \pm 0.3$ TgN yr[-1] and $0.5 \pm 0.3$ TgN yr[-1] (mean ±1 standard deviation) and between LGM and PI conditions (1500 CE) by $1.7 \pm 0.3$ and $0.7 \pm 0.3$ TgN yr[-1], respectively. Marine sources first dropped by ~ 0.5 TgN yr[-1] during the onset of HS1, when AMOC strongly decreased (McManus et al., 2004). Marine emissions remained minimal over the following millennium, but started to increase around 16 ka BP and recovered to LGM levels around 15 ka BP in the course of HS1, while AMOC remained

constantly low (McManus et al., 2004). This pattern of recovery during cold stadials has been observed before (Schilt et al., 2013) and tentatively attributed to a slow recovery of $N_2O$ production in subsurface waters as also suggested by marine nitrogen cycle modeling (Schmittner and Galbraith, 2008). Our quantitative emission reconstruction shows now for the first time the complete evolution of marine (and terrestrial) $N_2O$ emissions over the entire HS1 interval. The isotope results demonstrate that the millennial decrease at the beginning of HS1 and the later increase in atmospheric $N_2O$ during HS1 are

indeed caused by marine emissions and may be linked to the substantial ocean reorganization that is thought to have determined the early deglacial rise in atmospheric $CO_2$ (Schmitt et al., 2012;Tschumi et al., 2011;Jaccard et al., 2016). In contrast, terrestrial $N_2O$ emissions remained approximately invariant until the start of the B/A warm period in the northern



hemisphere. Then, land emissions rapidly increased at the start of the B/A, declined again during the YD, and peaked again after a rapid rise into the Preboral, consistent with earlier results for the 16 to 10 ka BP interval (Schilt et al., 2014). Also marine emission generally start to increase during the B/A warming, however, not as rapidly as terrestrial emissions and the main marine increase is delayed by a few hundred years relative to the onset of the rapid atmospheric $N_2O$ concentration

increase. A similar pattern is found for the rapid warming at the end of the YD. Note also that while terrestrial emissions remained relatively low over the entire course of the YD, marine emissions show a slow recovery that starts about 1000 years before the rapid $N_2O$ increase at the end of the YD into the Preboreal.

Compared to the drastic emission changes over the deglaciation, the trends in terrestrial $N_2O$ emissions are comparably small

during the Holocene. Our deconvolution suggests only a very small increase in terrestrial emissions over the first half of the Holocene. Consistently, the decline in atmospheric $N_2O$ concentrations in the early Holocene is attributed to a reduction in marine emissions. The 10 ppb higher $N_2O$ concentrations in the later Holocene are mainly attributed to an increase in terrestrial emissions which occurred predominantly in the time interval from about 6.5 to 5 ka BP. Marine emissions show only slightly increased values after 6 ka BP interrupted by a decline in marine emissions in the time interval 2.5-0.5 ka BP

(Fig. 7). However, the uncertainty in our reconstructed emission anomalies based on $N_2O$ and $\delta^{15}N(N_2O)$ during the Holocene is as large as the reconstructed emission changes (Fig. 7, solid and grey dashed lines), leaving some doubt on the significance of this reconstructed ocean emission anomaly. Looking at the $\delta^{18}O(N_2O)$ record in Fig. 3, which was not used in the deconvolution, $\delta^{18}O(N_2O)$ values are about 1 ‰ higher after 2.5 ka BP than before which - taken at face value - would contradict reduced $N_2O$ emissions by marine sources. A similar caveat applies to the millennial variability in marine and

terrestrial emission over the Holocene. The well-resolved and precise reconstruction of atmospheric $N_2O$ concentrations (Fig. 3) shows significant maxima around 4.5 and 2.5 ka BP and less pronounced maxima around 9.0, 7.3 and at 5.8 ka BP as well as minima at 8.3, 6.5, 3.6 and 1.4 ka BP, documenting significant millennial changes in total $N_2O$ emissions. However, the nitrogen isotope-based reconstruction of terrestrial and marine emissions does not yet allow for an unambiguous attribution of these changes to one source category. It may be tempting to attribute these millennial $N_2O$ emission changes

over the Holocene to changes in the AMOC strength or variations in the position of the ITCZ and the monsoonal rain belt (for example for the 8.2 ka cold event in Greenland (Spahni et al., 2003) or apparent large-scale climate changes in the course of the Holocene (Wanner et al., 2011)). However, no consistent connection of $N_2O$ emission maxima or minima with global climate changes can be recognized and no quantitative attribution to terrestrial or marine sources is possible at this point. To this end, higher resolution isotope records and further improvements in precision of the $\delta^{15}N(N_2O)$ analyses will be

necessary to better constrain small changes in the $N_2O$ budget over the Holocene.

Interestingly, our new ice core isotope data permit novel insights in the response time scales of terrestrial and marine $N_2O$ emissions during periods of rapid climate change. Using our box model deconvolution approach, we can in principle resolve





rapid emission changes which are otherwise hidden in the atmospheric $N_2O$ concentration record by the relatively long atmospheric lifetime (see methods). However, the ice core record presents a low-pass filtered version of the atmospheric changes due to the bubble enclosure process, which generally leads also to a documented response in the ice core record which is delayed and smoothed relative to the atmosphere. Moreover, as input data for the deconvolution we use the Monte

Carlo spline estimate of the $N_2O$ and $\delta^{15}N(N_2O)$ records, which adds additional low pass filtering to the ice core record. As outlined in section 2.2.2, the latter leads to an erroneously early response in the emission reconstruction for rapid $N_2O$ changes. In line, the change point marking the increase in reconstructed terrestrial and marine emissions at the onset of the B/A warming appears to occur earlier than the rapid $CH_4$ increase marking the rapid warming in the gas archive, but this lead is essentially an artefact imposed by the spline filtered data used for the deconvolution and is not supported by the

unsmoothed $N_2O$ concentration record. In order to pinpoint the onset of rapid changes in total $N_2O$ emissions more exactly, the higher resolution concentration record from the NGRIP ice core (Schilt et al., 2013), where the bubble enclosure process leads to an effective resolution in the order of a few decades for the Holocene and less than a century for the Last Glacial Maximum (Schilt et al., 2013), may be used. Taking the NGRIP data after outlier removal (Schilt et al., 2013) at face value (see discussion on potential in situ production above), this record suggests that $N_2O$ concentrations, thus total $N_2O$

emissions, started to change synchronously within the data resolution with the $CH_4$ concentration measured on the same samples (Fig. 2). In particular, the NGRIP $N_2O$ record excludes any significant $N_2O$ emission increase prior to the onset of the $CH_4$ rise, where the latter is also synchronous within a few decades with the rapid warming in Greenland temperatures as evidenced in the thermal diffusion signal in $\delta^{15}N_2$ in the NGRIP ice core (Baumgartner et al., 2014).

The low pass-filtering also leads also to an underestimation of the true atmospheric dynamics in $N_2O$ concentration changes. Using our Monte Carlo $N_2O$ concentration record compiled from all ice cores (Fig. 2), we see that 90 % of the concentration increase at the end of HS-1 is achieved over a time scale of about 500 years. Using artificial emission data in forward box model runs and low-pass filtering the calculated atmospheric concentrations with a log normal age distribution with a mean age of 132 yr (as applicable for the atmospheric record compiled from all our ice cores) this time scale can be achieved by a

linear increase in terrestrial emission within about 200 yr. In contrast, an emission increase over 300 yr already requires an increase in the (bubble low-pass filtered) ice core record over 600 yr. As our ice core $N_2O$ compilation is also low-pass filtered by the applied spline approximation, the true increase in the ice core concentration record should be even faster than 500 yrs, implying that the terrestrial emission increase happened within less than 200 yr. However, based on the resolution of the available records to date we cannot constrain the time scale of the $N_2O$ emission increase to better than 200 yr. In

particular, we refrain from using the NGRIP concentration data, which show a more rapid increase in $N_2O$ (Fig. 2) at the onset of the B/A and the end of the YD, to constrain this time scale, as these data appear to be affected by in situ formation of $N_2O$ in the ice in the respective time intervals (see discussion above).



A similar picture in terms of the timing of the onset and time scale of N₂O increase as at the onset of the B/A emerges for terrestrial emissions accompanying the warming at the end of the YD. Again, the major N₂O concentration rise in our data compilation starts synchronously within uncertainty with CH₄ and the warming in Greenland, while peak N₂O concentrations are delayed by a few centuries relative to the CH₄ maximum and the temperature change documented in Greenland climate

records (Steffensen et al., 2008). Accordingly, this time evolution of N₂O (terrestrial) emission changes appears to be a common feature for rapid climate changes during the last termination and potentially also for all DO events. Due to the preceding long-term increase in marine N₂O emissions over the YD event, a clear acceleration of the increase in marine N₂O emissions (as found for the B/A warming) is difficult to discern during the rapid warming at the end of the YD. A major increase in marine emissions, however, is reconstructed at around 11.4 ka BP, which is delayed relative to the onset of the

YD-Preboreal warming and lasts for about 200 years.

In summary, the terrestrial N₂O emissions increase at the beginning of the B/A period and the end of the YD were realized within 200 years. A more rapid, decadal-scale increase in terrestrial N₂O emissions is possible, but cannot be confirmed by the available ice core information to date. In any case the terrestrial emissions increased much more rapidly than marine

emissions during rapid climate warming, indicating a longer response time of the ocean compared to the land biosphere as also suggested in a marine N₂O modeling study (Goldstein et al., 2003). Modelling studies also suggest that the main response of marine N₂O production and emissions is delayed by a few centuries relative to AMOC changes in idealized freshwater hosing experiments and even a longer delay is simulated until marine emission changes have reached their full amplitude (Schmittner and Galbraith, 2008;Battaglia et al., 2019).

**3.2.2 Sensitivity analyses**

To further evaluate our emission reconstruction, δ¹⁸O(N₂O) is carried as a tracer in the atmospheric box model and tropospheric δ¹⁸O(N₂O) changes are calculated based on the reconstructed land and ocean emissions. The mean results of this forward modeling suggest little change in tropospheric δ¹⁸O(N₂O) in agreement with the ice core δ¹⁸O(N₂O). Yet, the uncertainty range in projected δ¹⁸O(N₂O) is with about ±1.5 ‰ about two times larger than the analytical uncertainty of an

individual δ¹⁸O(N₂O) ice core measurement (Fig. 3). This is due to the relatively large uncertainties in the input data (mainly the isotopic source signatures) used in our Monte Carlo box model approach. The large model uncertainty in addition to the complex nature of the global cycle of δ¹⁸O, prevents any firm conclusions from δ¹⁸O(N₂O) data.

We further test the robustness of the Monte Carlo deconvolution approach to infer changes in global N₂O emissions using

sensitivity analyses (see Fig. 6). The results suggest a moderate sensitivity of inferred land and ocean N₂O emissions to plausible changes in the global mean δ¹⁵N of N₂O emissions from land and from the ocean but still well within the overall uncertainty ranges obtained from the Monte Carlo procedure. In other words, the sensitivity of inferred N₂O emissions to





plausible changes in $\delta^{15}N$ of the global land and ocean emissions is significantly smaller than the reconstructed glacial/interglacial and rapid changes. In particular, a large deglacial decrease in the land isotopic signature by 2 ‰ as suggested by lacustrine data (McLauchlan et al., 2013), requires a reduction in the glacial/interglacial land emission change by only 0.4 TgN yr⁻¹ compared to our standard scenario (and an equivalent increase in marine emissions). Moreover, while

the total glacial/interglacial increase may change (in opposite directions for marine and terrestrial emissions) the relative millennial variability seen in our records is not affected by these source scenarios.

Second, the influence of the prior assumption on the initial fraction of marine emissions relative to total $N_2O$ emissions is investigated. In the standard Monte Carlo setup, the marine contribution at the start of the deconvolution is uniformly varied

between 30 % and 58 % of total emissions (Table 2) following the most recent observation-constrained estimate (Battaglia and Joos, 2018a). For comparison, the same range would yield a preindustrial range of 3.3 to 6.6 TgN yr⁻¹ in marine $N_2O$ emissions. In two sensitivity tests, we investigate the influence of the prescribed initial range and vary the initial fraction of marine emissions between 25 % and 35 % (green line in Fig. 6C to E) and between 53 and 63 % (blue line in Fig. 6C to E) only. Assuming such strong deviations from the observation-based range of potential marine fractions, the calculated

terrestrial and marine emissions shift towards the edge of the $2\sigma$ error bands of the standard deconvolution (Fig. 6) and the glacial/interglacial increase in terrestrial $N_2O$ emissions is larger by about 0.6 TgN yr⁻¹ and smaller by about 0.3 TgN yr⁻¹ in the two sensitivity runs, respectively. However, the temporal evolution of relative changes in land and marine $N_2O$ emissions remains similar. In conclusion, the absolute magnitude of change in land and ocean $N_2O$ emissions is sensitive to the assumed split between marine and terrestrial $N_2O$ emissions which is still uncertain (Ciais et al., 2013), however, the relative

changes in the temporal evolution of marine and terrestrial emissions are not.

Finally, we varied the atmospheric lifetime over the deglacial period in an idealized scenario assuming an overall decrease in lifetime from 143 to 123 yr from the glacial to the Holocene. This change in lifetime causes a parallel increase in both land and ocean emissions by about 16 % which then lie close to the $2\sigma$ error of our standard deconvolution. Late Holocene

emission anomalies relative to 21 ka BP increase by about 0.6 Tg yr⁻¹ for both terrestrial and marine emissions. Again, the assumed scenario for past lifetime changes has little effect on the temporal evolution of relative emission changes.

In conclusion, the main features of our standard reconstruction such as the decrease and recovery of global marine $N_2O$ emissions during the HS1 and YD intervals and the rapid rise in global terrestrial emissions at the onset of the B/A and the

end of the YD are robust. The reconstructed long-term variations clearly exceed the uncertainty ranges revealed by our Monte Carlo analysis where both parameters and ice core data were varied within their uncertainties. The relative constancy of terrestrial $N_2O$ emissions and the slow decline in marine emissions after the Preboreal appears to be robust as well as the finding of constant terrestrial emissions during the LGM and HS1 and the overall deglacial increase in both global marine and global terrestrial $N_2O$ emissions to the atmosphere. However, the exact magnitude of the millennial scale fluctuations



and of the deglacial increase in terrestrial and marine emissions depends on the assumed initial ratio of terrestrial to marine $N_2O$ emissions and on potential changes in the atmospheric lifetime of $N_2O$.

## 4. Discussion

### 4.1 Reconstructed $N_2O$ emission changes

We followed the approach by Schilt et al. (2014) and deconvolved the ice core records of $N_2O$ and $\delta^{15}N(N_2O)$ to quantify changes in terrestrial and marine $N_2O$ emissions. These emission records serve as benchmarks for ocean and land models simulating $N_2O$ emissions to the atmosphere. The amplitude of the reconstructed deglacial change in marine versus terrestrial emissions depends sensitively on the assumed initial ratio of land versus ocean emissions, however the relative changes in land and ocean emission represent a robust feature of our box model deconvolution. In our standard scenario, the fraction of

marine emissions has been set to a central value of 44 % and allowed to vary between 30 and 58 %, consistent with most recent evidence (Battaglia and Joos, 2018a). The central value corresponds to preindustrial emissions of 4.6 TgN yr$^{-1}$ from the ocean and of 5.9 TgN yr$^{-1}$ from the land.

The temporal resolution of reconstructed $N_2O$ atmospheric concentration and marine and terrestrial emission changes is

limited by the smoothing properties of the ice archive. Specifically, the transfer of atmospheric air to the firn layer and the enclosure of firn air into the ice leads to an age distribution of the air conserved in the ice and finally measured. This enclosure process acts like an asymmetric low-pass filter with a strong bias to old ages of the air enclosed, removing fast variations in $N_2O$ and $N_2O$ isotopes. In addition, for our deconvolution the ice core measurements are smoothed with a spline that acts like a (symmetric) low-pass filter to produce a continuous record from the large, but limited number of samples, and

to account for the small, but noticeable scatter in the ice core data due to measurement and archive uncertainties. Taken together, these limitations restrict the temporal resolution of change that can be unveiled by our deconvolution. Nevertheless, we can firmly state that the terrestrial $N_2O$ emission changes at the onset of the B/A and the end of the Younger Dryas started synchronously with the increase in $CH_4$ and occurred within maximum two centuries, but potentially even faster.

### 4.2 Marine $N_2O$ emission changes

Marine $N_2O$ emissions are controlled by the amount of organic matter re-mineralized and oxygen ($O_2$) concentrations (Suntharalingam and Sarmiento, 2000;Battaglia and Joos, 2018a). $N_2O$ production by nitrification increases with substrate availability and with decreasing $O_2$ concentration. The overall increase in marine $N_2O$ emission over the termination is as such consistent with paleoceanographic evidence showing an oxygen depletion across the deglaciation in the upper ocean (Jaccard and Galbraith, 2012;Moffitt et al., 2015), whereas changes in organic matter export as inferred from ocean

sediments varied regionally (Kohfeld et al., 2005) and overall organic matter preservation was much higher during the LGM than the Holocene in low and mid-latitudes (Cartapanis et al., 2016). The largest volumetric expansion of oxygen-depleted



water masses is reconstructed for the onset of the B/A, when oxygen concentrations decreased throughout the Indo-Pacific above 2,500 m (Jaccard et al., 2014) and oxygen minimum zones expanded world-wide (Moffitt et al., 2015). High resolution records from the North Pacific clearly show that hypoxic events occurred in response to the fast warming into the B/A and at the end of the YD and were connected to local productivity feedbacks (Praetorius et al., 2015). In summary,

upper ocean oxygen depletion appears as the most likely explanation for the marine $N_2O$ emission increase from 16 to 13.5 ka BP, perhaps partly counteracted by a potential decrease in organic matter export and remineralization.

Marine $N_2O$ emissions show a very sharp drop around 17.4 ka BP at the onset of the Heinrich Stadial 1 (HS1; 17.4 to 14.6 ka BP) and reduced emissions for the next ~1500 years. This decreased $N_2O$ emission occurred together with the onset of the

deglacial $CO_2$ rise (Monnin et al., 2001;Marcott et al., 2014) and a concomitant equally sharp decrease in $\delta^{13}C(CO_2)$ (Schmitt et al., 2012), an increase in the deposition of biogenic opal in Southern Ocean sediments indicative of Southern Ocean upwelling (Anderson et al., 2009), a drop in atmospheric $\Delta^{14}C$ (Southon et al., 2012;Reimer et al., 2013), and an increase in Southern Ocean oxygenation (Jaccard et al., 2016), as well as the occurrence of ice-rafted debris in Iberian Margin sediments (Bard et al., 2000). Apart from the ice-rafted debris, these changes have been explained by an

intensification of Southern Ocean upwelling bringing carbon and silica rich, $\delta^{13}C$ and $\Delta^{14}C$ depleted waters from the deep ocean to the surface (Schmitt et al., 2012), and are consistent with the multi-tracer relationships found in dynamic ocean-biogeochemistry model simulations (Tschumi et al., 2011). The drop in marine $N_2O$ emissions at the onset and the low emissions during parts of the HS1 may accordingly be linked to an increase in Southern Ocean ventilation and ocean oxygen concentration during the first part of HS1.

Interestingly, both reconstructed marine $N_2O$ emissions and atmospheric $CO_2$ show a trend change within HS1. Marine $N_2O$ emissions remained minimal within the interval from about 17.4 ka BP to 16 ka BP, while atmospheric $CO_2$ increased by about 30 ppm during this period. Around 16 ka BP, marine $N_2O$ emissions started to rise and the growth trend in atmospheric $CO_2$ declined. At the same time a minimum in $\delta^{13}C(CO_2)$ is reached and followed by a small increase (Schmitt

et al., 2012). The change in $N_2O$ emissions and the trend reversal in $\delta^{13}C(CO_2)$ may point to a change in the mechanisms governing the rise in atmospheric $CO_2$ which, in turn, lead to a slower $CO_2$ growth rate. This is consistent with the suggestion that Southern Ocean ventilation changes were mainly responsible for the early $CO_2$ rise (Tschumi et al., 2011).

Paleo $N_2O$ modelling studies are sparse, but three studies present partly conflicting results for YD-type climatic events. All

studies report results from dynamic ocean models forced with freshwater hosing in the North Atlantic to provoke a collapse of the Atlantic Meridional Overturning Circulation and cooling in the North Atlantic realm. In a zonally averaged ocean model, Goldstein et al. (2003) Simulated reduced marine emissions in response to reduced Atlantic overturning and a respective increase during AMOC resumption. This effect can explain about half of the reconstructed atmospheric $N_2O$





decrease/increase during the AMOC collapse/resumption. These authors proposed that terrestrial $N_2O$ emission changes must be responsible for the remaining atmospheric $N_2O$ variation and, in particular, caused the rapid increase in atmospheric $N_2O$ at the end of the YD. A recent nitrogen cycle ocean model study by Battaglia et al. (2019) also shows a reduction/increase of marine $N_2O$ emissions of only 0.8 TgN yr$^{-1}$, during AMOC collapse/resumption, which is not sufficient

to explain the full amplitude of atmospheric $N_2O$ changes observed in the ice core record. In contrast, in the ocean model by Schmittner and Galbraith (2008) the variation in marine productivity is able to explain the full $N_2O$ concentration change observed in ice cores in response to AMOC variations. Our novel quantitative emission reconstructions support the findings by Goldstein et al. (2003) and Battaglia et al (2018). The reconstructions indeed show a slower recovery of marine $N_2O$ emissions and a much faster increase in terrestrial emissions towards the end of the YD, a $N_2O$ emission scenario already

suggested by Schilt et al. (2010b).

While the centennial to millennial scale oscillations in terrestrial and marine emissions over the Holocene cannot be interpreted with certainty due to the still limited sampling resolution and precision of the Holocene $\delta^{15}N(N_2O)$ data, the longer term decrease in the marine emissions over the first half of the Holocene appears significant and synchronous to a

reconstructed slow increase in AMOC and subsurface water ventilation in the Atlantic Ocean (McManus et al., 2004;Waelbroeck et al., 2018) and a change in Southern Ocean bottom water properties (Jaccard et al., 2016). Thus, the processes invoked during the AMOC resumption at the end of HS1 and the end of the YD leading to a century-scale increase in marine $N_2O$ emissions, cannot explain the early Holocene decrease of ocean sources. Moffitt et al. (2015) (their Table 5) show that oxic conditions expanded between 10 and 4 ka BP in the Pacific eastern upwelling systems. An increase in oxygen

minimum zone ventilation from the early to mid-Holocene is reconstructed for the Arabian Sea (Kessarkar et al., 2018) and has been related to an intensified winter monsoon and winter mixing arising from cooling in the North Atlantic region (Reichart et al., 1998). We speculate that the gradual AMOC intensification (McManus et al., 2004), accompanying the demise of the northern hemisphere ice sheets at a time when interglacial temperatures and sea ice extent were already established, lead to an improvement in oxygenation in the low-latitude thermocline of the Atlantic and, in concert with

orbital forcing and monsoonal wind changes, in the Pacific and Indian upwelling systems that may have led to the reconstructed decrease in marine $N_2O$ emissions over the first half of the Holocene.

Generally, the difference in the time scales associated with the reconstructed marine and terrestrial emission changes during the YD and over the entire termination is consistent with the notion that terrestrial $N_2O$ emissions can react quickly to

climate change. In contrast, marine emission changes materialize on longer centennial to up to millennial time scales associated with surface-to-deep tracer transport and with physical and biogeochemical adjustment processes, such as changes in ocean circulation, export productivity, oxygen concentrations and the marine $N_2O$ inventory as also indicated by multi-millennial global warming projections of marine oxygen and $N_2O$ (Battaglia and Joos, 2018b).



### 4.3 Terrestrial N₂O emission changes

Terrestrial emissions show a 40 % increase from the Last Glacial Maximum to the late Holocene period. Most of the deglacial increase was realized in two fast and large steps. These occurred at the onset of the B/A and at the end of the YD, when reconstructed terrestrial $N_2O$ emissions changed by about 1 TgN yr$^{-1}$. These large changes were realized within a period of two centuries, and possibly faster, considering the low pass filtering effects of the ice archive and in the input data used for the deconvolution. The B/A onset and the end of the YD are linked to rapid, decadal-scale and widespread northern hemisphere warming, to shifts in the Intertropical Convergence Zone (ITCZ) and precipitation patterns, and to fast changes in atmospheric $CH_4$ (Baumgartner et al., 2014;Rhodes et al., 2015).  The latter are indicative of terrestrial methane emissions (Bock et al., 2017;Spahni et al., 2011) mainly in the tropics.

Reconstructed terrestrial $N_2O$ emissions changed relatively little during the Holocene. This is in contrast to the large reconstructed changes in land biosphere carbon stocks and in atmospheric $CO_2$ and $CH_4$ (Elsig et al., 2009;Flückiger et al., 2002). Ice core data of $CO_2$ and $\delta^{13}C(CO_2)$ imply an increase in terrestrial C storage of 290±36 PgC during the early Holocene until about 6.5 ka BP followed by a smaller release thereafter (Elsig et al., 2009). At the same time our terrestrial $N_2O$ emission reconstruction suggests only a minor increase from 11 until 6.5 ka BP, a modest increase in terrestrial emissions of up to 0.2 TgN yr$^{-1}$ from 6.5 to 5 ka BP and a rather constant overall level thereafter. The significant increase in carbon storage in the early Holocene suggests that N supply was more than sufficient to support both generally high $N_2O$ emissions and an increase in carbon inventory in boreal peatlands (Yu et al., 2010) and possibly other regions (Stocker et al., 2017).

The apparent decoupling of significant net terrestrial carbon storage from $N_2O$ emissions in the early Holocene and an increase in $N_2O$ emissions from 6.5 to 5 ka BP, when $CO_2$ concentrations stayed relatively constant, implies that a linear relationship between terrestrial net carbon storage and $N_2O$ production is not an adequate picture for relative changes in nitrogen turnover in soils. While it is not possible to pinpoint the responsible regional terrestrial $N_2O$ sources from the globally integrated information stored in atmospheric $N_2O$ and $\delta^{15}N(N_2O)$ in ice cores, it is worthwhile to note that the time interval from 11 to 6.5 ka BP was characterized by a significant decrease in sea level rise rates (Lambeck et al., 2014), the disappearance of the remnants of glacial northern hemisphere ice sheets and therefore most likely increased soil formation in high northern latitudes. Moreover, the Holocene was characterized by a strong decline in NH summer insolation starting around 7 ka BP and continuing over the entire Holocene with significant impacts on the position of the ITCZ and monsoonal precipitation (Wang et al., 2005;Marcott et al., 2013), however, globally integrated terrestrial $N_2O$ emissions stayed relatively constant after 5 ka BP. Thus, global nitrogen turnover in soils and the connected $N_2O$ production cannot be attributed to a controlling single factor (such as precipitation changes) or a specific region of the globe, but most likely





average over competing changes in different regions throughout the globe (see also Joos et al. (this issue) for a detailed terrestrial model study of the controlling processes and the contributing source areas).

## 5 Summary and Conclusions

We reconstructed the evolution of the stable isotopes $\delta^{15}N$ and $\delta^{18}O$ of $N_2O$ for the last 28,000 years. Our $N_2O$ concentration
record is seamlessly linked with the Law Dome ice and firn $N_2O$ data and to instrumental $N_2O$ measurements of tropospheric background air (MacFarling Meure et al., 2006). The age scales of the records are aligned with the absolutely counted GICC05 age scale (Rasmussen et al., 2006) using high resolution methane measurements from Antarctica and Greenland. This leads to particularly small age scale uncertainties around periods of rapid climate and methane changes, namely, the onset of the B/A (14.6 to 12.8 ka BP) northern hemisphere warm period and the beginning and the end of the YD (12.8 to
11.7 ka BP) northern hemisphere cold period.

For the last 21,000 years, where (isotopic) $N_2O$ ice core data is available in sufficient resolution and from several ice cores from both polar regions, the $N_2O$ concentration and $\delta^{15}N(N_2O)$ records allowed for a quantitative reconstruction of terrestrial and marine $N_2O$ emissions. These provide important benchmarks for fully coupled, $N_2O$ and isotope-enabled Earth System
Models (ESM). Particularly interesting features to investigate are the $N_2O$ depression during Heinrich Stadial I (HS1; 17.4 to 14.6 ka BP), the rapid increase in $N_2O$ to preindustrial levels at the onset of the B/A and the YD fluctuation, as well as the Holocene evolution of terrestrial and marine $N_2O$ emissions.

Our records show that both terrestrial and marine $N_2O$ emissions took part in the rapid $N_2O$ increases during rapid deglacial
warmings connected to the resumption of the AMOC as already shown earlier (Schilt et al., 2014). For the first time we are able to show that this is also true for the general glacial/interglacial $N_2O$ increase, while some decoupling is suggested for long-term terrestrial and marine emission trends during the Holocene. In particular, the strong deglacial increase in terrestrial $N_2O$ emissions, indicative of an enhanced nitrogen turnover in soils, implies that nitrogen availability was not a limiting factor of vegetation growth on the longer time scales and supports the idea of significant biological nitrogen fixation over
time (see Joos et al., this issue). Reconstructed terrestrial $N_2O$ emissions changed by up to 1 TgN yr$^{-1}$ and within less than two centuries at the onset of the NH warming events around 14.6 and 11.7 ka BP, while marine emissions changed more sluggishly to ocean reorganizations accompanying the rapid warming events during the deglacial. These time scales are also relevant for 21$^{st}$ century climate projections and much longer than accessible in typical laboratory or field experiments. Taken at face value, our results suggest that the terrestrial nitrogen cycle will also adjust on the global scale in the coming
decades towards meeting N demand to support carbon uptake by plants under raising $CO_2$ as projected in some dynamic vegetation models (Stocker et al., 2013), while marine $N_2O$ feedbacks to future global warming induced changes in AMOC will only fully unfold on time scales of centuries in the future.





*Acknowledgements.* Long-term financial support of this research by the Swiss National Science Foundation (grant no. 200020_172506 and 200020_172476) is gratefully acknowledged. Part of the research leading to these results has received funding from the European Research Council under the European Union's Seventh Framework Programme (FP/2007-2013) /

5   ERC Advanced Grant Agreement no. 226172 (MATRICs) awarded to HF. This work is also a contribution to the European Project for Ice Coring in Antarctica (EPICA), the Talos Dome Ice Core Project (TALDICE) and the North Greenland Ice Core Project (NGRIP). EPICA is a joint European Science Foundation/European Commission (EC) scientific programme, funded by the EC and by national contributions from Belgium, Denmark, France, Germany, Italy, The Netherlands, Norway, Sweden, Switzerland and the UK. The main logistic support was provided by IPEV and PNRA (at Dome C) and AWI (at

10  Dronning Maud Land). TALDICE is a joint European programme led by Italy and is funded by national contributions from Italy, France, Germany, Switzerland and the United Kingdom. The main logistical support was provided by PNRA at Talos Dome. NGRIP is directed and organised by the Department of Geophysics at the Niels Bohr Institute for Astronomy, Physics and Geophysics, University of Copenhagen. It is supported by funding agencies in Denmark, Belgium, France, Germany, Iceland, Japan, Sweden, Switzerland and the USA. This is EPICA publication no. XX and TALDICE publication no. YY.

*Data availability.* Data presented in this study will be available at the NOAA paleoclimate data base

*Competing interests.* The authors declare that they have no conflict of interest.




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





**Table 1: Measurement precision (±1 standard deviation) estimates of N₂O isotope ice core data measured at the University of Bern in this study.**

| Samples/Isotope | system I (Bock et al., 2014) | | system II (Schmitt et al., 2014) | |
| --- | --- | --- | --- | --- |
| | $\delta^{15}N(N_2O)$ (‰) | $\delta^{18}O(N_2O)$ (‰) | $\delta^{15}N(N_2O)$ (‰) | $\delta^{18}O(N_2O)$ (‰) |
| NGRIP 2013 | 0.2 to 0.3 | 0.5 to 0.7 | 0.3 | 0.4 |
| EDML 2013 | 0.2 to 0.3 | 0.5 to 0.7 | 0.3 | 0.4 |
| EDML 2014 | 0.2 to 0.4 | 0.5 to 0.8 | 0.3 | 0.4 |
| TALDICE | Not measured | Not measured | 0.3 | 0.4 |

**Table 2: Parameter ranges used in the atmospheric two-box model for the calculation of terrestrial and marine N₂O emissions.**

| Parameter | Range | Unit | Distribution | Reference |
| --- | --- | --- | --- | --- |
| Initial fraction of marine source | 0.30 – 0.58 | - | Uniform | (Battaglia and Joos, 2018a) |
| $\delta^{15}N(N_2O)$ terrestrial | -34 – 2 | ‰ | Uniform | (Schilt et al., 2014) |
| $\delta^{15}N(N_2O)$ marine | 4 – 10 | ‰ | Uniform | (Frame et al., 2014) |
| $\delta^{18}O(N_2O)$ terrestrial | 3 – 45 | ‰ | Uniform | (Park et al., 2011;Schilt et al., 2014) |
| $\delta^{18}O(N_2O)$ marine | 38 – 58 | ‰ | Uniform | (Frame et al., 2014) |
| Lifetime | 123 ± 9 | yr | Gaussian | (Prather et al., 2015) |
| Stratospheric fractionation constant ($^{15}N$) | -16.8 ± 1.6 | ‰ | Gaussian | (Röckmann et al., 2001) |
| Stratospheric fractionation constant ($^{18}O$) | -13.8 ± 2.0 | ‰ | Gaussian | (Röckmann et al., 2001) |
| Number of moles in atmosphere | 1.77 | $10^{20}$ mol | Constant | (Röckmann et al., 2003) |
| Stratospheric fraction of total atmosphere | 0.15 | - | Constant | (Röckmann et al., 2003) |
| Exchange rate troposphere/stratosphere | 4.11 – 6.63 | $10^{17}$ kg yr$^{-1}$ | Uniform | (Ishijima et al., 2007) |



**Figure 1: Compilation of new and published N₂O records over the last 28 kyr including identified artefacts due to in situ production of N₂O. (A) N₂O concentration from this study and published records (Sowers et al., 2003;Bernard et al., 2006;Flückiger et al., 2002;MacFarling Meure et al., 2006;Park et al., 2012;Schilt et al., 2010b;Schmitt et al., 2014) (B) δ¹⁵N(N₂O) and (C) δ¹⁸O(N₂O) from this study and published records (Sowers et al., 2003;Bernard et al., 2006;Park et al., 2012;Schmitt et al., 2014;Schilt et al., 2014). The solid lines in A-C represent Monte Carlo Average smoothing splines with a cutoff period of 2000 yr prior to 16 ka BP and 700 yr afterwards. The grey shaded envelopes (±2 sigma) represent the Monte Carlo based uncertainties of the splines. Samples identified to be subject to in situ formation are indicated by red circles; extreme values outside the plotting range are indicated by arrows and provided in small orange boxes.**





**Figure 2: Methane synchronization of ice core records. (A) Existing methane (CH$_4$) concentration records for NGRIP and EDML (Baumgartner et al., 2012), and TALDICE (Schilt et al., 2010b) shown on the original AICC2012 gas age scale. The fast climate variations indicated by the NGRIP δ$^{18}$O$_{ice}$ record (B) (Rasmussen et al., 2006), were used as tie points for CH$_4$ changes in the NGRIP and TALDICE ice cores (mid-points of the CH$_4$ changes indicated by vertical dashed red lines). The onset of the CH$_4$ increase at the end of HS1 is indicated by the blue vertical dashed line (see text). A similar approach had been taken by (Schilt et al., 2010b) by synchronizing the fast changes in the Taylor Glacier CH$_4$ data with the corresponding changes in the WAIS Divide deep ice-core CH$_4$ data on an updated version of the WDC06A-7 timescale. The synchronised CH$_4$ records (C) are then used to create a common gas age scale for N$_2$O data on the GICC05 age scale (D) (Rasmussen et al., 2006).**



**Figure 3: Greenhouse gas records over the last 21 kyr after removing $N_2O$ samples affected by in situ formation. (A) $CO_2$ concentration compilation (Bereiter et al., 2015), (B) $CH_4$ concentration (Loulergue et al., 2008;Rhodes et al., 2015), (C) $N_2O$ concentration as compiled in this study, (D) $\delta^{15}N(N_2O)$ and (E) $\delta^{18}O(N_2O)$ from (Schilt et al., 2014) and this study. The solid lines in C-E represent Monte Carlo Average smoothing splines with a cutoff period of 2000 yr prior to 16 ka BP and 700 yr afterwards. The grey shaded envelopes represent of the Monte Carlo based uncertainties (±2 sigma) of the splines. The dashed envelope in panel E, depicts the ±2 sigma range of the forward modeled atmospheric $\delta^{18}O(N_2O)$ signal expected from the terrestrial and marine emissions reconstructed using $N_2O$ concentrations and $\delta^{15}N(N_2O)$ (see also Figure 7).**





**Figure 4: Late Holocene evolution of (A) N₂O concentrations from this study and published records (Bernard et al., 2006;Flückiger et al., 2002;MacFarling Meure et al., 2006;Park et al., 2012;Schilt et al., 2010b;Schmitt et al., 2014;Rubino et al., 2018;Prokopiou et al., 2018) (B) δ¹⁵N(N₂O) and (C) δ¹⁸O(N₂O) from this study and published records (Bernard et al., 2006;Park et al., 2012;Schmitt et al., 2014;Prokopiou et al., 2018), connecting the ice core records and the independent reconstructions using firn air and the Cape Grim air archive (Park et al., 2012).**




**Figure 5: Deconvolution of artificial ice core data for a hypothetical 50 yr emission ramp-up of (i) terrestrial emissions, (ii) marine emissions and (iii) terrestrial and marine emissions in a fixed ratio. (A) assumed terrestrial and/or marine emission increases for scenarios (i) to (iii). The increase of total emissions is the same in all cases, however the contributions of terrestrial and marine**
5  **sources differ in the three scenarios. $f_m$ denotes the change in the fraction of marine to total emissions during these three runs. (B) calculated $N_2O$ concentration before (dashed black line) and after low-pass filtering with a log-normal gas age distribution with a mean gas age of 132 yr mimicking the bubble enclosure process (solid black line). The dotted black line represents the artificial ice core data after applying a spline approximation with a cutoff period of 700yr. (C) calculated $\delta^{15}N(N_2O)$ before (dashed colored lines) and after low-pass filtering (solid colored lines) with the log-normal gas age distribution and after applying a spline**
10  **approximation with cutoff period of 700 yr (dotted colored lines), (D) deconvolution results using the data in B and C showing terrestrial (solid colored lines) and marine (dashed colored lines) emissions. Also plotted for comparison are the real emissions reconstructed from the ice core data at the onset of the B/A warming (grey lines in A-D), where the ice core results are plotted on the model time scale assuming that the well-defined onset in the ice core $CH_4$ increase (vertical blue dashed line in Figure 2) is synchronous to the onset of the $N_2O$ emission increase in the artificial data.**



**Figure 6: Results from sensitivity analyses to test the robustness of N₂O emissions inferred by deconvolving the ice core N₂O and δ¹⁵N(N₂O) records. Reconstructed, potential changes in oxygen (A) and nitrogen (B) isotopic source signatures of precursor material relative to 21 ka BP. Inferred changes in terrestrial (C) and marine (D) N₂O emissions anomalies relative to 21 ka BP, E) atmospheric δ¹⁸O(N₂O) from ice cores (black dots) and inferred from the terrestrial and marine emissions in panel C and D. Emission anomalies of the standard deconvolution in C-E are shown in black including their ± 1 standard deviation uncertainty (grey shading). Sensitivity setups refer to runs with a lowered (green) and elevated (blue) marine source fraction range at the start of the runs, temporal terrestrial source signature changes derived from model data (purple line) and from lacustrine sediments (red line), and using a time dependent atmospheric lifetime (yellow) as described in detail in section 2.2.3. In both source signature scenarios (purple and red line), marine source signatures are varied according to evidence from marine sediment records.**





**Figure 7: Reconstructed global terrestrial (A) and marine (B) N₂O emission changes:** Mean reconstructed terrestrial and marine N₂O emissions are shown by the solid green and blue lines, respectively, together with their uncertainty estimate (±1 sigma; colored shaded area) based on 500 accepted runs calculated using the Monte Carlo atmospheric two-box model (Schilt et al., 2014) and ice records of N₂O concentration and δ¹⁵N(N₂O). (C) mean reconstructed marine fraction $f_m$ of total emissions (solid black line) and its 1-sigma uncertainty (grey shaded area). y-axes on the right indicate the terrestrial and marine emission anomalies relative to the LGM (21 ka BP). The dashed grey lines around the mean anomalies indicate the ±1 sigma uncertainty of the reconstructed anomalies. Note that the uncertainties of the anomalies are smaller than those of the total fluxes, as the choice of the isotopic source signatures in the Monte Carlo approach systematically shifts total emission fluxes in an individual run to higher or lower values but not the anomalies. Thus, we are able to reconstruct changes in the N₂O emissions more precisely than their overall level using our approach.