# Peer review of "N2O changes from the Last Glacial Maximum to the preindustrial - part I"

_Biogeosciences, 2019_

## Referee Comment (RC1) · Anonymous Referee #1 · 29 May 2019

This manuscript by Fischer et al. presents N2O gas observations including isotopes in ice cores from the Last Glacial Maximum to preindustrial. They build upon previous work using a Monte Carlo two-box model to interpret the data and estimate changes to terrestrial and marine emissions. The authors conclude that both terrestrial and marine emissions must have increased over the deglaciation, with the terrestrial contribution likely to be about a factor of 2 larger than marine, and discuss uncertainties. A more robust result from the analysis is the temporal dynamics that indicate two sharp increases in terrestrial emissions at the beginning of the B/A and end of the YD, while

marine emissions are driven by longer millennial fluctuations that appear to be linked to North Atlantic climate/AMOC.

Overall I find this to be an excellent study that gives new quantitative insights to N2O emissions over the last 28,000 years. The simple box model is a simple but suitable framework to estimate N2O emissions. However, I have a couple of issues regarding the model estimates that should be addressed before I would recommend publication.

Major Comment: Uncertainty of total terrestrial vs. marine emission increases over the deglaciation

In the abstract, the uncertainty levels on the deglacial increase of N2O emissions are small at 0.3 Tg N yr-1, which gives the impression that there is a high degree of certainty on the relative contribution of terrestrial and marine emissions, which is one of the most important results of this study. After reading the discussion in the text, this seems much more uncertain. For example on page 20, lines 1-2: "... the deglacial increase in terrestrial and marine emissions depends on the assumed initial ratio of terrestrial to marine N2O emissions...". This implies that the contribution of terrestrial vs. marine emissions over the deglaciation is determined from a modern model estimate that could have been much different in LGM conditions and thus introduces large uncertainties. My guess is that the temporal dynamics of the model fit to the observations prevent a large deviation from this imposed initial assumption, which could be more clearly described.

In any case, the selection of the uncertainty levels (±0.3 Tg N yr-1) should be specifically discussed. I guess it comes from the uncertainty in anomalies which is only described in one sentence in the caption of Figure 7, and thus I do not fully understand. It yields an uncertainty level of 0 at the LGM, i.e. assumes that the imposed initial condition based on a modern model is also correct in the LGM as I understand it. I would have thought the uncertainty should be at its largest levels during the LGM and deglaciation since there is additional uncertainty regarding how end-member isotopic

values may have changed. In Figure 6, the low-biased and high-biased marine fraction sensitivity simulations suggest much higher uncertainties in emissions that seems more realistic since this is a key assumption/uncertainty driving the model estimate.

Minor comments:

Page 1, line 17: "show" should be something like "estimate" or "suggest" since that number is a model prediction, not an observation

Page 8, lines 30-32: "varied between -34 and +2 ‰ for the global terrestrial emissions and between +4 and +10 ‰ for global marine emissions"

What are the actual best-fit isotopic end-member values used in the model simulations? Given these wide ranges, I am curious how many plausible scenarios exist that can still explain the observations. Since the presented emission change scenarios are mostly consistent, I assume the range if values in the "accepted runs calculated by the Monte Carlo atmospheric two-box model" is quite narrow. Is that correct? In general, I wonder how useful the isotopic constraint is given these large ranges of end-member values.

Page 19, lines 17-18: "However, the temporal evolution of relative changes in land marine N2O emissions remains similar".

This is where I get a bit confused. Figure 6b shows large differences in the relative marine emissions for the different marine fraction scenarios. I would say that a scenario that remains near peak deglacial levels across the Holocene (high-biased fm) has a different temporal evolution to one that drops back to LGM values during the Holocene (low-biased fm), even if the smaller wiggles correspond.

Discussion: Previous work by Schilt et al., 2014 suggested an equal contribution from terrestrial and marine increases over the deglaciation, whereas this study suggest a larger contribution from terrestrial emissions relative to marine. What part of the data and/or model analysis differed in this study that led to this change?

---

## Referee Comment (RC2) · Anonymous Referee #2 · 10 Jun 2019

Fischer et al. in this manuscript present a compilation of N2O and its isotopic (both N and O isotopes) data for the last 12,000 years (28,000 years with less resolution data) from ice-core records from both Greenland and Antarctica by combining new high-resolution analysis with previously available measurements. They then use N2O concentration and N isotope data to provide a quantitative reconstruction of terrestrial and marine N2O emission history.

They find that N2O emissions from land and ocean increased during the last deglaciation, closely linked with climate warming and ocean circulation. Also, land emissions Printer-friendly version

responded abruptly to Northern Hemisphere climate warming at the onsets of BOA and the Holocene, in particular to monsoon and ITCZ shifts.

The compilation and interpretation are comprehensive and provides fundamental data sets for understanding carbon-nitrogen cycle processes, especially in Earth system models. The manuscript is well organized, and the writing is clear, despite complex data sets and technical issues involved and required discussions.

I do not have major concerns about the manuscript. I think that the manuscript can be accepted for publication after considering the following specific comments, mostly minor.

Page 1 Line 21: Add "Almost", or "Up to" or "More than" before "90% of these large stpe increases". Do not start a sentence with a number.

L21: change from "within maximum two centuries" to "within two centuries at maximum"?

L25: change to "in reconstructed marine N2O emission of 0.4 TgN yr-1" L26: change "suggesting" to "suggests"

L27: change "however" to "but"?

L28: change "which" to "that"

Page 2 L5-6: the discussion on land and marine processes is a bit confusing. Or change "where nitrification" to "but nitrification"? The reference of Battaglia and Joo 2017 should be 2018a?

L7: change "in line with" to "similar to", as the new estimate technically is outside the range in IPCC as cited, 9.5 vs. 9.0. In any case, "in line with" is unclear in wording.

Page 3 L18: change "Two hundred two" to "A total of 202 ice samples" (also delete "core"?)

BGD
L22: change "13 samples" to "A subset of 13 samples", as not to start a sentence with a number.

Page 7 L3: add "," before "which does not..."

L26: change "Also" to "Also," L26: delete one "in this interval". Also, the meaning "this interval" is unclear so maybe just repeat "late Holocene" as this is the first sentence in a new paragraph.

Page 10 L27: delete "," after "Note"

Page 11 L13: add ;" before "however,"

Page 12 L4: the equation: I don't think you should add units in the equation. They are awkward and confusing. If for absolute clarity, you could use a, b and c for three coefficients in the equation and then define their values and units, such as "a = 4.3266 per mil" (also, indicate the unit used for MAT and MAP—and d15N for completeness).

Page 13 L27: change "between 26-18 ka BP" to "at 26-18 ka BP", or "between 26 and 19 ka BP" (the former is concise and crisp, so preferable in situation like this).

Page 14 L1: change "deglacial" to "deglaciation"

Page 23 L30: change ", however," to "; however,"

Page 36 L4: delete "," after "Rasmussen et al, 2006)"

Page 37 L7: delete "," after panel E"

BGD

---

## Author Comment (AC1) · 28 Jun 2019

Please see the attached supplementary file for our reply to the review comments.

Please also note the supplement to this comment:
https://www.biogeosciences-discuss.net/bg-2019-117/bg-2019-117-AC1-supplement.pdf

---

## Author Response (AR1)

**Replies to the reviews**

Anonymous Referee #1

This manuscript by Fischer et al. presents N2O gas observations including isotopes in ice cores from the Last Glacial Maximum to preindustrial. They build upon previous work using a Monte Carlo two-box model to interpret the data and estimate changes to terrestrial and marine emissions. The authors conclude that both terrestrial and marine emissions must have increased over the deglaciation, with the terrestrial contribution likely to be about a factor of 2 larger than marine, and discuss uncertainties. A more robust result from the analysis is the temporal dynamics that indicate two sharp increases in terrestrial emissions at the beginning of the B/A and end of the YD, whilemarine emissions are driven by longer millennial fluctuations that appear to be linked to North Atlantic climate/AMOC.

Overall I find this to be an excellent study that gives new quantitative insights to N2O emissions over the last 28,000 years. The simple box model is a simple but suitable framework to estimate N2O emissions. However, I have a couple of issues regarding the model estimates that should be addressed before I would recommend publication. Major Comment: Uncertainty of total terrestrial vs. marine emission increases over the deglaciation

We thank the reviewer for his/her overall positive assessment of our work and will try to meet the points of criticisms as outlined below. Note that in addition to the textual changes outlined below we will provide improved versions of Figures 5 and 7.

In the abstract, the uncertainty levels on the deglacial increase of N2O emissions are small at 0.3 Tg N yr-1, which gives the impression that there is a high degree of certainty on the relative contribution of terrestrial and marine emissions, which is one of the most important results of this study. After reading the discussion in the text, this seems much more uncertain. For example on page 20, lines 1-2: ": : : the deglacial increase in terrestrial and marine emissions depends on the assumed initial ratio of terrestrial to marine N2O emissions: : :". This implies that the contribution of terrestrial vs. marine emissions over the deglaciation is determined from a modern model estimate that could have been much different in LGM conditions and thus introduces large uncertainties. My guess is that the temporal dynamics of the model fit to the observations prevent a large deviation from this imposed initial assumption, which could be more clearly described.

We thank the reviewer for this comment as it illustrates that we did not explain sufficiently in the text the difference in the uncertainty of the absolute level of the marine and terrestrial emission fluxes on the one side and the flux anomalies relative to the LGM level on the other. The prior uncertainty is much larger (colored error bar in Figure 7) than the latter (dashed lines in Figure 7). The reason for this is that the absolute level of marine and terrestrial emissions is strongly dependent on the individual choice of the isotopic source signatures in each Monte Carlo run. This choice, thus, also largely determines the marine fraction. However, in the course of the transition and the Holocene, the isotopic source signature remains the same in each Monte Carlo run and, thus, individual runs are offset from each other in the marine and terrestrial emission fluxes (in opposite directions), but this offset remains essentially the same over the entire last 21 kyr (BTW, this also implies that the marine fraction of each individual run does not change strongly over the last 21 kyr). To illustrate that explain this in more detail in the manuscript and add three individual example runs in Figure 7 that clearly show the offsets, while the anomalies relative to the LGM are very similar. In particular we add on page 15:

*The $N_2O$ and $\delta^{15}N(N_2O)$ records allow us to disentangle changes in global terrestrial and global marine $N_2O$ emissions to the atmosphere since the LGM. To this end we used the two-box model deconvolution of the $N_2O$ and $\delta^{15}N(N_2O)$ records described in section 2.2 to determine marine and terrestrial $N_2O$ emissions and their uncertainty (Fig. 7) over the last 21 kyr. The uncertainty (colored shading in Figure 7) of the absolute emissions (left y-axes) is relatively large, reflecting the large range of possible isotopic source signatures for marine and terrestrial emissions accepted in the box model runs, which spreads essentially over the entire allowed range of terrestrial and marine source signatures. Together with the constraint on the marine emission fraction, this determines the absolute level of terrestrial and marine emissions in each of our accepted Monte Carlo runs (see Figure 7 for three examples of individual runs (grey lines)). Note that these individual runs are systematically offset from each other dependent on the choice of the isotopic source signature, but that the anomalies relative to the LGM level of all runs are very similar. Thus using our Monte Carlo box model deconvolution with randomly chosen but temporal constant isotopic source signatures, we can quantify emission anomalies relative to the LGM level much more precisely than the absolute emission level using or deconvolution.*

[Figure]

(new Figure 7)

5   In any case, the selection of the uncertainty levels (0.3 Tg N yr-1) should be specifically discussed. I guess it
    comes from the uncertainty in anomalies which is only described in one sentence in the caption of Figure 7, and
    thus I do not fully understand. It yields an uncertainty level of 0 at the LGM, i.e. assumes that the imposed initial
    condition based on a modern model is also correct in the LGM as I understand it. I would have thought the
    uncertainty should be at its largest levels during the LGM and deglaciation since there is additional uncertainty
10  regarding how end-member isotopic values may have changed. In Figure 6, the low-biased and high-biased
    marine fraction sensitivity simulations suggest much higher uncertainties in emissions that seems more realistic
    since this is a key assumption/uncertainty driving the model estimate.

The reviewer is correct that the uncertainties provided in the abstract refer to the uncertainties in the anomalies relative to the LGM level. As mentioned above this uncertainty is much smaller than the one of the absolute emissions levels. We will stress this more clearly in the revised manuscript.

5  Please note that the results of the sensitivity studies using scenarios of systematic changes in the isotopic source signature over the transition (scenario 1 and 2) lie within the 1 sigma uncertainty of our standard runs, where for the latter the isotopic source signatures are kept constant. Also the high biased marine fraction scenario lies within the 1 sigma uncertainty of our standard runs and the low biased scenario still lies within the 2-sigma uncertainty. In particular the low-biased scenario should be regarded as an extreme scenario (see also below).

10  Accordingly, the provided 1 sigma uncertainty of the anomalies in land and marine emission fluxes relative to the LGM represent a realistic error estimate of the temporal changes.

As correctly pointed out by the reviewer , the low biased scenario implies very small absolute LGM-Holocene changes in marine emissions (and larger changes for terrestrial emissions). However, at the same time this

15  scenario requires a substantial decline of marine $N_2O$ emission over the course of the Holocene, which is hard to explain for a period of relatively constant climate. While a low marine fraction as required in this low biased run is entirely possible for the LGM, this run implies also a very low marine fraction during preindustrial times, which is at the  bottom end of the potential marine fraction estimates by Battaglia and Joos (2018b) for that time. A lower glacial marine fraction and higher fraction during the late Holocene requires to change the isotopic

20  source signature over time similar to what is shown in the sensitivity scenario 1 (depleted terrestrial emissions in the course of the termination).

We expand the discussion on this topic in the revised manuscript. In particular we will add on page 19 of the revised manuscript:

*"Second, the influence of the prior assumption on the initial fraction of marine emissions relative to total $N_2O$ emissions is investigated. In the standard Monte Carlo setup, the marine contribution at the start of the deconvolution is uniformly varied between 30 % and 58 % of total emissions (equivalent to a range of 3.3 to 6.6 TgN yr$^{-1}$ in preindustrial marine $N_2O$ emissions), following the most recent observation-constrained estimate (Battaglia and Joos, 2018b) with a best-guess*

30  *estimate of 43 % very close to the mean preindustrial value in our reconstruction (Figure 7). In two sensitivity tests, we investigate the influence of the prescribed initial range and vary the initial fraction of marine emissions between 25 % and*

*35 % (low-biased scenario, green line in Fig. 6C to E) and between 53 and 63 % (high-biased scenario, blue line in Fig. 6C to E) only. Assuming such strong deviations from the best estimate for the marine fraction, the Holocene emission anomalies relative to the LGM level are shifted by about +1σ of the standard runs towards higher marine emissions in the high-biased scenario and by -2σ of the standards runs towards lower marine emissions in the low-biased scenario. In fact, the latter sensitivity run does show only a very small change in marine emissions between the LGM and the late Holocene. However, while a low marine fraction during the LGM as required in this run is possible, the preindustrial marine fraction in this run is lower than the best guess estimate by Battaglia and Joos (2018b), thus this run is likely underestimating the Holocene increase in marine emissions. To reconcile the required increase in marine fraction from the LGM to the late Holocene and our isotopic constraints asks for a significant shift in the isotopic source signatures, similar to what is observed in source signature scenario 1. We conclude, that the results of our sensitivity studies overall support the robustness of our results and that the standard deviation of the emission anomalies relative to the LGM level in the standard runs provides a representative uncertainty estimate for possible emission changes. While we stress that the absolute magnitude of land and ocean $N_2O$ emissions is sensitive to selected isotopic source signature and the assumed split between marine and terrestrial $N_2O$ emissions, the relative changes in the temporal evolution of marine and terrestrial emissions are much less affected by this choice."*

Minor comments:

Page 1, line 17: "show" should be something like "estimate" or "suggest" since that number is a model prediction, not an observation

*"Our reconstruction indicates"*

Page 8, lines 30-32: "varied between -34 and +2 ‰ for the global terrestrial emissions and between +4 and +10 ‰ for global marine emissions"

What are the actual best-fit isotopic end-member values used in the model simulations? Given these wide ranges, I am curious how many plausible scenarios exist that can still explain the observations. Since the presented emission change scenarios are mostly consistent, I assume the range if values in the "accepted runs calculated by the Monte Carlo atmospheric two-box model" is quite narrow. Is that correct? In general, I wonder how useful the isotopic constraint is given these large ranges of end-member values.

The isotopic signatures in the accepted Monte Carlo runs show a rather wide distribution. For the land emissions this distribution covers the interval from -5 to -34 permille and has a relatively wide Gaussian shape, where more accepted runs are found in the land signature range between -10 and -23 permille. For the marine emissions the distribution of accepted runs is quasi uniform and covers the entire allowed isotopic range from +4 to +10 permille. Thus, the range of accepted values is actually quite wide. This is also the reason why the uncertainty of our estimates of the absolute emission fluxes are relative large (while the uncertainties of the anomalies relative to the LGM are not). This will be discussed in the revised manuscript (see first comment above).

Note that a higher frequency of accepted runs for a certain (land) isotopic signature implies only that it is easier for the Monte Carlo model to find a solution within the error limits, but does imply that these specific isotopic signatures are more likely than others.

Page 19, lines 17-18: "However, the temporal evolution of relative changes in land marine N2O emissions remains similar".

sentence changed to: "However, the temporal evolution of anomalies in both land and marine $N_2O$ emissions relative to the LGM values remains similar."

This is where I get a bit confused. Figure 6b shows large differences in the relative marine emissions for the different marine fraction scenarios. I would say that a scenario that remains near peak deglacial levels across the Holocene (high-biased fm) has a different temporal evolution to one that drops back to LGM values during the Holocene (low-biased fm), even if the smaller wiggles correspond.

The high-biased and low-biased sensitivity runs should be regarded as extreme scenarios as the preindustrial marine fraction implied in these scenarios does not agree with the best estimate by Battaglia and Joos (see discussion above). Note again, that the marine fraction in each individual run is largely determined by the source signatures and does not change largely over the last 21 kyr. Thus, the late Holocene values of the marine emissions in the low-biased scenario are too low compared to our current knowledge and the land emissions therefore too high as long as source signatures are not allowed to change over the transition. Thus, the green line

in Figure 6 illustrates the systematic effects of the marine fraction constraint but cannot be regarded as realistic scenarios for the Holocene. We will mention that in the revised manuscript as outlined in comment 2 above.

Discussion: Previous work by Schilt et al., 2014 suggested an equal contribution from terrestrial and marine increases over the deglaciation, whereas this study suggest a larger contribution from terrestrial emissions relative to marine. What part of the data and/or model analysis differed in this study that led to this change?

The major difference is the point in time against which anomalies are calculated. In the paper by Schilt the data only covered the time period from 16 kyr to 10 kyr BP. In fact, for this time period our deconvolution is largely the same as in Schilt, with minor modifications by a few additional data points and by adjusting our best guess marine fraction interval to latest results (Battaglia and Joos, 2018a). However, Schilt et al. could only provide anomalies relative to 16 kyr and not relative to the LGM, thus, did not provide a true deglacial estimate. In fact, the marine emissions show a clear minimum at 16 kyr BP, i.e., in Heinrich Stadial 1, explaining most of the difference in the marine emission anomaly estimate between Schilt et al 2014 and our value. We will mention in the revised manuscript on page 15 that:

*This drop into HS1 also explains the apparently higher marine emission change estimate in Schilt et al. (2014), who, due to the limited data availability at that time, provided an emission anomaly relative to the value at 16 kyr instead of 21 kyr BP.*

Anonymous Referee #2

Fischer et al. in this manuscript present a compilation of N2O and its isotopic (both N and O isotopes) data for the last 12,000 years (28,000 years with less resolution data) from ice-core records from both Greenland and Antarctica by combining new high-resolution analysis with previously available measurements. They then use N2O concentration and N isotope data to provide a quantitative reconstruction of terrestrial and marine N2O emission history.

They find that N2O emissions from land and ocean increased during the last deglaciation, closely linked with climate warming and ocean circulation. Also, land emissions responded abruptly to Northern Hemisphere climate warming at the onsets of BOA and the Holocene, in particular to monsoon and ITCZ shifts.

The compilation and interpretation are comprehensive and provides fundamental data sets for understanding carbon-nitrogen cycle processes, especially in Earth system models. The manuscript is well organized, and the writing is clear, despite complex data sets and technical issues involved and required discussions.

I do not have major concerns about the manuscript. I think that the manuscript can be accepted for publication after considering the following specific comments, mostly minor.

Page 1 Line 21: Add "Almost", or "Up to" or "More than" before "90% of these large step increases". Do not start a sentence with a number.

done

L21: change from "within maximum two centuries" to "within two centuries at maximum"?

done

L25: change to "in reconstructed marine N2O emission of 0.4 TgN yr-1" L26: change "suggesting" to "suggests"

done

L27: change "however" to "but"?

done

L28: change "which" to "that"

"The latter is currently"

Page 2 L5-6: the discussion on land and marine processes is a bit confusing. Or change "where nitrification" to "but nitrification"?

done

The reference of Battaglia and Joos 2017 should be 2018a?

5  done

L7: change "in line with" to "similar to", as the new estimate technically is outside the range in IPCC as cited, 9.5 vs. 9.0. In any case, "in line with" is unclear in wording.

10  The statement in the text is correct. The 10.5 TgN/yr refer to the total emission, while the 6.6. Tg/yr in the IPCC refer to terrestrial emissions, i.e. about 60%. We changed the wording to "very similar to"

Page 3 L18: change "Two hundred two" to "A total of 202 ice samples" (also delete "core"?)

15  done

L22: change "13 samples" to "A subset of 13 samples", as not to start a sentence with a number.

done

Page 7 L3: add "," before "which does not: : :"

done

25  L26: change "Also" to "Also,"

done

L26: delete one "in this interval". Also, the meaning "this interval" is unclear so maybe just repeat "late

30  Holocene" as this is the first sentence in a new paragraph.

done

Page 10 L27: delete "," after "Note"

done

Page 11 L13: add ;" before "however,"

done

Page 12 L4: the equation: I don't think you should add units in the equation. They are awkward and confusing. If for absolute clarity, you could use a, b and c for three coefficients in the equation and then define their values and units, such as "a = 4.3266 per mil" (also, indicate the unit used for MAT and MAPâ˜A ˘Tand d15N for completeness).

we prefer to leave the units in the equation to avoid confusion. We added the units for MAT and MAP. in the text.

Page 13 L27: change "between 26-18 ka BP" to "at 26-18 ka BP", or "between 26 and 19 ka BP" (the former is concise and crisp, so preferable in situation like this).

done

Page 14 L1: change "deglacial" to "deglaciation"

done

Page 23 L30: change ", however," to "; however,"

done

Page 36 L4: delete "," after "Rasmussen et al, 2006)"

done

5   Page 37 L7: delete "," after panel E"

done

[revised manuscript text omitted]
₂O emissions and their uncertainty (Fig. 7) over the last 21 kyr. The uncertainty (colored shading in Figure 7) of the absolute emissions (left y-axes) is relatively large, reflecting the large range of possible isotopic source signatures for marine and terrestrial emissions accepted in the box model runs, which spreads essentially over the entire allowed range of terrestrial and marine source signatures. Together with the constraint on the marine emission fraction, this determines the absolute level of terrestrial and marine emissions in each of our accepted Monte Carlo runs (see Figure 7 for three examples of individual runs (grey lines)). Note that these individual runs are systematically offset from each other dependent on the choice of the isotopic source signature, but that the anomalies relative to the LGM level of all runs are very similar. Thus using our Monte Carlo box model deconvolution with randomly chosen but temporal constant isotopic source signatures, we can quantify emission anomalies relative to the LGM level much more precisely than the absolute emission level using or deconvolution.

Reconstructed terrestrial and marine N₂O emissions increased between the LGM (21 ka BP) and full interglacial conditions (7 ka BP) by $1.5 \pm 0.3$ TgN yr$^{-1}$ and $0.5 \pm 0.3$ TgN yr$^{-1}$ (mean ±1 standard deviation) and between LGM and PI conditions (1500 CE) by $1.7 \pm 0.3$ and $0.7 \pm 0.3$ TgN yr$^{-1}$, respectively. Marine sources first dropped by ~ 0.5 TgN yr$^{-1}$ during the onset of HS1, when AMOC strongly decreased (McManus et al., 2004). This drop into HS1 also explains the apparently higher marine emission change estimate in Schilt et al. (2014), who, due to the limited data availability at that time, provided an

[revised manuscript text omitted]
 $\delta^{18}O(N_2O)$ in agreement with the ice core $\delta^{18}O(N_2O)$. Yet, the uncertainty range in projected $\delta^{18}O(N_2O)$ is with about ±1.5 ‰ about two times larger than the analytical

uncertainty of an individual $\delta^{18}O(N_2O)$ ice core measurement (Fig. 3). This is due to the relatively wide range of possible input data (mainly the isotopic source signatures) used in our Monte Carlo box model approach. The large model uncertainty in addition to the complex nature of the global cycle of $\delta^{18}O$, prevents any firm conclusions from $\delta^{18}O(N_2O)$ data.

We further test the robustness of the Monte Carlo deconvolution approach to infer changes in global $N_2O$ emissions using sensitivity analyses (see Fig. 6). The results suggest a moderate sensitivity of inferred land and ocean $N_2O$ emissions to plausible changes in the global mean $\delta^{15}N$ of $N_2O$ emissions from land and from the ocean but still well within the overall uncertainty ranges obtained from the Monte Carlo procedure. In other words, the sensitivity of inferred $N_2O$ emissions to
10  plausible changes in $\delta^{15}N$ of the global land and ocean emissions is smaller than the reconstructed glacial/interglacial and rapid changes. In particular, a large deglacial decrease in the land isotopic signature by 2 ‰ as suggested by lacustrine data (McLauchlan et al., 2013), requires a reduction in land emission change relative to the LGM value  by only about 0.4 TgN yr$^{-1}$ compared to our standard scenario (and an equivalent increase in marine emissions), i.e., very close to the 1σ uncertainty of our standard runs. Moreover, while the total glacial/interglacial increase may change (in opposite directions for marine
15  and terrestrial emissions) the relative millennial variability seen in our records is not affected by these source scenarios.

Second, the influence of the prior assumption on the initial fraction of marine emissions relative to total $N_2O$ emissions is investigated. In the standard Monte Carlo setup, the marine contribution at the start of the deconvolution is uniformly varied between 30 % and 58 % of total emissions (equivalent to a range of 3.3 to 6.6 TgN yr$^{-1}$ in preindustrial marine $N_2O$
20  emissions), following the most recent observation-constrained estimate (Battaglia and Joos, 2018b) with a best-guess estimate of 43 % very close to the mean preindustrial value in our reconstruction (Figure 7). In two sensitivity tests, we investigate the influence of the prescribed initial range and vary the initial fraction of marine emissions between 25 % and 35 % (low-biased scenario, green line in Fig. 6C to E) and between 53 and 63 % (high-biased scenario, blue line in Fig. 6C to E) only. Assuming such strong deviations from the best estimate for the marine fraction, the Holocene emission anomalies
25  relative to the LGM level are shifted by about +1σ of the standard runs towards higher marine emissions in the high-biased scenario and by -2σ of the standards runs towards lower marine emissions in the low-biased scenario. In fact, the latter sensitivity run does show only a very small change in marine emissions between the LGM and the late Holocene. However, while a low marine fraction during the LGM as required in this run is possible, the preindustrial marine fraction in this run is lower than the best guess estimate by Battaglia and Joos (2018b), thus this run is likely underestimating the Holocene
30  increase in marine emissions. To reconcile the required increase in marine fraction from the LGM to the late Holocene and our isotopic constraints asks for a significant shift in the isotopic source signatures, similar to what is observed in source signature scenario 1. We conclude, that the results of our sensitivity studies overall support the robustness of our results and that the standard deviation of the emission anomalies relative to the LGM level in the standard runs provides a representative

uncertainty estimate for possible emission changes. While we stress that the absolute magnitude of land and ocean $N_2O$ emissions is sensitive to selected isotopic source signature and the assumed split between marine and terrestrial $N_2O$ emissions, the relative changes in the temporal evolution of marine and terrestrial emissions are much less affected by this choice.

Finally, we varied the atmospheric lifetime over the deglacial period in an idealized scenario assuming an overall decrease in lifetime from 143 to 123 yr from the glacial to the Holocene. This change in lifetime causes a parallel increase in both land and ocean emissions by about 16 %. Late Holocene emission anomalies relative to 21 ka BP increase by about 0.6 Tg yr$^{-1}$ for both terrestrial and marine emissions. Again, the assumed scenario for past lifetime changes alters the absolute amplitude of
10 emission anomalies relative to the LGM, but has little effect on the temporal evolution of the emissions.

In conclusion, the main features of our standard reconstruction such as the decrease and recovery of global marine $N_2O$ emissions during the HS1 and YD intervals and the rapid rise in global terrestrial emissions at the onset of the B/A and the end of the YD are robust. The reconstructed emission anomalies relative to the LGM level clearly exceed the uncertainty
15 ranges revealed by our Monte Carlo analysis where both parameters and ice core data were varied within their uncertainties. The rather constant terrestrial $N_2O$ emissions in the Holocene and the slow decline in marine emissions after the Preboreal appears to be robust as well as the finding of constant terrestrial emissions during the LGM and HS1 and the overall deglacial increase in both global marine and global terrestrial $N_2O$ emissions to the atmosphere.

**4. Discussion**

20 **4.1 Reconstructed $N_2O$ emission changes**

We followed the approach by Schilt et al. (2014) and deconvolved the ice core records of $N_2O$ and $\delta^{15}N(N_2O)$ to quantify changes in terrestrial and marine $N_2O$ emissions. These emission records serve as benchmarks for ocean and land models simulating $N_2O$ emissions to the atmosphere. The total emission changes are only a function of atmospheric lifetime changes, however, the split of total emission in marine versus terrestrial emissions depends on isotopic source signatures and
25 the assumed initial ratio of land versus ocean emissions. However within reasonable bounds of the marine emission fraction the changes in land and ocean emission relative to the LGM level 
[revised manuscript text omitted]

Enting, I.G., 1987. On the use of smoothing splines to filter $CO_2$ data. J Geophys Res 92, 10977-10984.

Erhardt, T., Capron, E., Rasmussen, S.O., Schüpbach, S., Bigler, M., Adolphi, F., Fischer, H., 2019. Decadal-scale progression of the onset of Dansgaard–Oeschger warming events. Clim. Past 15, 811-825.

Etheridge, D.M., Pearman, G.I., de Silva, F., 1988. Atmospheric trace-gas variations as revealed by air trapped in an ice core from Law Dome, Antarctica. Ann Glac 10, 28-33.

Flückiger, J., Blunier, T., Stauffer, B., Chappellaz, J., Spahni, R., Kawamura, K., Schwander, J., Stocker, T.F., Dahl-Jensen, D., 2004. $N_2O$ and $CH_4$ variations during the last glacial epoch: Insight into global processes. Global Biogechemical Cycles 18, doi:10.1029/2003GB002122.

Flückiger, J., Dällenbach, A., Blunier, T., Stauffer, B., Stocker, T.F., Raynaud, D., Barnola, J.M., 1999. Variations in atmospheric $N_2O$ concentration during abrupt climatic changes. Science 285, 227-230.

Flückiger, J., Monnin, E., Stauffer, B., Schwander, J., Stocker, T.F., Chappellaz, J., Raynaud, D., Barnola, J.M., 2002. High-resolution Holocene $N_2O$ ice core record and its relationship with $CH_4$ and $CO_2$. Glob Biogeochem Cyc 16, 1010, doi:1010.1029/2001GB001417.

Fourteau, K., Faïn, X., Martinerie, P., Landais, A., Ekaykin, A.A., Lipenkov, V.Y., Chappellaz, J., 2017. Analytical constraints on layered gas trapping and smoothing of atmospheric variability in ice under low-accumulation conditions. Clim. Past 13, 1815-1830.

Frame, C.H., Deal, E., Nevison, C.D., Casciotti, K.L., 2014. $N_2O$ production in the eastern South Atlantic: analysis of $N_2O$ stable isotopic and concentration data. Glob Biogeochem Cyc, 2013GB004790.

Francey, R.J., Allison, C.E., Etheridge, D.M., Trudinger, C.M., Enting, I.G., Leuenberger, M., Langenfelds, R.L., Michel, E., Steele, L.P., 1999. A 1000-year high precision record of $\delta^{13}C$ in atmospheric $CO_2$. Tellus B 51, 170-193.

Galbraith, E.D., Kienast, M., 2013. The acceleration of oceanic denitrification during deglacial warming. Nature Geosci 6, 579-584.

5 Goldstein, B., Joos, F., Stocker, T.F., 2003. A modeling study of oceanic nitrous oxide during the Younger Dryas cold period. Geophys Res Let 30, doi: 10.1029/2002GL016418.

Henry, L.G., McManus, J.F., Curry, W.B., Roberts, N.L., Piotrowski, A.M., Keigwin, L.D., 2016. North Atlantic ocean circulation and abrupt climate change during the last glaciation. Science 353, 470.

Ishijima, K., Sugawara, S., Kawamura, K., Hashida, G., Morimoto, S., Murayama, S., Aoki, S., Nakazawa, T., 2007. 10 Temporal variations of the atmospheric nitrous oxide concentration and its $\delta^{15}N$ and $\delta^{18}O$ for the latter half of the 20th century reconstructed from firn air analyses. J Geophys Res-Atmos 112.

Jaccard, S.L., Galbraith, E.D., 2012. Large climate-driven changes of oceanic oxygen concentrations during the last deglaciation. Nature Geoscience 5, 151-156.

Jaccard, S.L., Galbraith, E.D., Frölicher, T.L., Gruber, N., 2014. Ocean (de)oxygenation across the last deglaciation: Insights 15 for the future. Oceanography 27, 26-35.

Jaccard, S.L., Galbraith, E.D., Martínez-García, A., Anderson, R.F., 2016. Covariation of deep Southern Ocean oxygenation and atmospheric $CO_2$ through the last ice age. Nature 530, 207-210.

Joos, F., Spahni, R., Stocker, B.D., Lienert, S., Müller, J., Fischer, H., Schmitt, J., Prentice, I.C., Otto-Bliesner, B., Liu, Z., 2019. $N_2O$ 
[revised manuscript text omitted]


[revised manuscript text omitted]